# Demystifying Embedding Spaces using Large Language Models

**Guy Tennenholtz,**\* **Yinlam Chow, Chih-Wei Hsu, Jihwan Jeong, Lior Shani,**
**Azamat Tulepbergenov, Deepak Ramachandran, Martin Mladenov, Craig Boutilier**

Google Research

## Abstract

Embeddings have become a pivotal means to represent complex, multi-faceted information about entities, concepts, and relationships in a condensed and useful format. Nevertheless, they often preclude direct interpretation. While downstream tasks make use of these compressed representations, meaningful interpretation usually requires visualization using dimensionality reduction or specialized machine learning interpretability methods. This paper addresses the challenge of making such embeddings more interpretable and broadly useful, by employing large language models (LLMs) to directly interact with embeddings – transforming abstract vectors into understandable narratives. By injecting embeddings into LLMs, we enable querying and exploration of complex embedding data. We demonstrate our approach on a variety of diverse tasks, including: enhancing concept activation vectors (CAVs), communicating novel embedded entities, and decoding user preferences in recommender systems. Our work couples the immense information potential of embeddings with the interpretative power of LLMs.

## 1 Introduction

The success of deep learning has brought forth a paradigm shift in knowledge representation through the concept of *embeddings*—dense vector representations that capture high-dimensional information about entities, concepts, or relationships in a compact and useful format. Embeddings are ubiquitous, finding application in natural language processing (Mikolov et al., 2013; Devlin et al., 2018), recommender systems (Rendle, 2010), protein sequence modeling (Rives et al., 2021), and more. These embeddings are invaluable, capturing nuanced relationships and semantic structure in data which traditional machine learning (ML) approaches often miss. Nevertheless, understanding these abstract representations remains challenging.

By design, the structure and underlying information carried by an embedding is heavily adapted to the idiosyncrasies of the downstream task, posing a substantial challenge to its interpretation and manipulation. Previous work on ML interpretability offers various task-independent means to interpret embeddings, including dimensionality reduction techniques (e.g., *t-SNE* (Van der Maaten & Hinton, 2008), *UMAP* (McInnes et al., 2018)) or *concept activation vectors (CAVs)* (Kim et al., 2018). While very useful, these techniques are fairly narrow in scope.

As an alternative to such interpretability methods, suppose one could engage with embeddings using natural language to query information not directly expressed by the name or description of the underlying entity/concept. Perhaps one could even extract information from the embedding representations of *non-existent or hypothetical* entities. Consider, for instance, an embedding space representing items from an online commerce site, trained using a large corpus of user ratings or purchases, reviews, and other multi-faceted data sources. The embedding representation of an item may *implicitly* embody intricate details about its quality, usability, design, customer satisfaction, etc. Moreover, suppose one wanted to understand the properties of a *hypothetical* item at a specific point in the embedding space, say, if we predicted a potential market for such an item. Since no such item (or description) exists, asking a conventional language model to describe this embedding point would

---

\*Correspondence to: `guytenn@gmail.com`

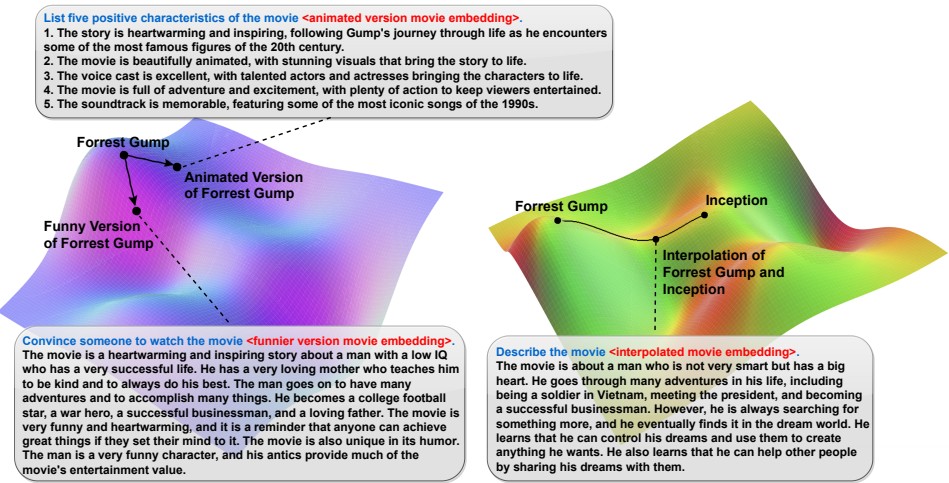

Figure 1: Decoded outputs of our model, ELM, for hypothetical movie embeddings. Text in blue and red correspond to the textual and domain embedding portions of the prompt to ELM. Text in black corresponds to the output of ELM.

be futile. However, a large language model (LLM) trained to *interpret the embedding representation itself* could handle such a task (see Fig. 1 for an illustration).

This paper introduces a novel framework to interpret *domain embeddings*[1] by leveraging the power of LLMs (Devlin et al., 2018; Liu et al., 2021; Google et al., 2023). Our method seamlessly introduces embeddings into LLMs by training *adapter layers* to map domain embedding vectors into the token-level embedding space of an LLM, which in turn allows one to treat these vectors as token-level encodings of the entities or concepts they represent. We train an LLM on a collection of tasks designed to facilitate the robust, generalizable interpretation of vectors in the domain embedding space. Our approach allows us to engage in a direct "dialogue" about these vectors, to query the LLM with intricate embedding data, and tease out narratives and insights from these dense vectors.

Our contributions are as follows. First, we formulate the problem of interpreting embeddings using LLMs. We then propose the *Embedding Language Model (ELM)*, a novel language model framework which, using trained adapters, can accept domain embedding vectors as parts of its textual input sequence to allow interpretation of continuous domain embeddings using natural language. We also develop a training methodology to fine-tune pretrained LLMs for domain-embedding interpretation. Finally, we test ELM by constructing 25 training tasks to allow interpretation of movie and user embeddings derived from the MovieLens 25M dataset (Harper & Konstan, 2015). We demonstrate our trained ELM on a variety of problems, including: generalizing CAVs as an interpretability method, describing hypothetical embedded entities, and interpreting user embeddings in a recommender systems by generating preference profiles. Our work bridges the gap between the rich data representations of domain embeddings and the expressive capabilities of LLMs.

## 2 INTEGRATING DOMAIN EMBEDDINGS WITH LARGE LANGUAGE MODELS

We assume a *domain embedding* $E_D : \mathbb{V} \to \mathcal{W}$ which maps entities in some vocabulary $\mathbb{V}$ (e.g., a set of known users and items) into a latent space $\mathcal{W} \subseteq \mathbb{R}^n$, equipped with a distance metric $d : \mathcal{W} \times \mathcal{W} \mapsto \mathbb{R}_+$. The embedding vector $E_D(v)$ of any $v \in \mathbb{V}$ is typically a representation used in some downstream task, and is trained to encompass the latent features of $v$ necessary for this task. For example, in a recommender system, *collaborative filtering* (CF) (Salakhutdinov & Mnih, 2007; Koren et al., 2011) is often used to generate embedding vectors of users and items s.t. the dot product (or cosine similarity) of $E_D(u)$ and $E_D(i)$ is predictive of user $u$'s affinity for item $i$. In

---

[1]We use the term "domain embedding" to differentiate the external embeddings to be interpreted from those captured by an LLM at various layers of the network.

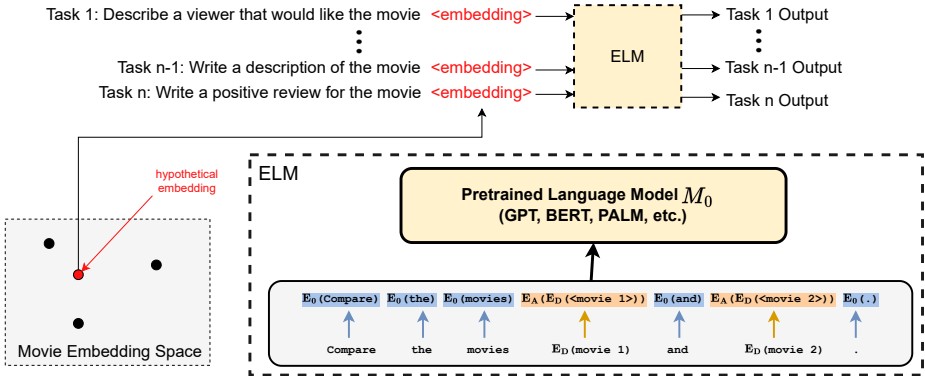

Figure 2: We train ELM using $n$ tasks that incorporate embeddings as tokens in language. Specifically, to incorporate the domain embedding space $\mathcal{W}$, we enhance the pretrained LLM $\mathcal{M}$ with an adapter model $E_A : \mathcal{W} \mapsto \mathcal{Z}$ to create a new language model $\mathcal{M}_{\text{ELM}}$, ensuring tokens and embeddings are mapped to a shared space. While $E_0$ projects language tokens to $\mathcal{Z}$, the adapter $E_A$ learns to project domain embedding vectors $E_D$ from embedding space $\mathcal{W}$ to the same space, $\mathcal{Z}$. The resulting sequence is input to a pretrained language model $M_0$, which can be further fine-tuned (see Sec. 2).

Table 1: Glossary

| | | | |
|---|---|---|---|
| $\mathcal{W}$ | Domain embedding space | $d$ | Domain space metric |
| $\mathcal{Z}$ | Token (language) embedding space | $\mathcal{X}$ | Token space |
| $E_0$ | LLM embedding | $\mathcal{M}$ | Pretrained LLM |
| $E_A$ | Adapter ($\mathcal{W} \mapsto \mathcal{Z}$) | $\mathcal{M}_{\text{ELM}}$ | Embedding Language Model |
| $E_D$ | Domain embedding | $M_0$ | LLM excluding input embedding layer |

image classification, $E_D$ might generate embedding representations useful for object detection (Zou et al., 2023), while in search/information retrieval, $E_D$ might embed both queries and documents in $\mathcal{W}$ to measure a document's relevance to a query (Zuccon et al., 2015; Luan et al., 2021).

We assume a pretrained LLM, $\mathcal{M} : \mathcal{X}^H \mapsto \mathcal{X}^H$, which maps (length $H$) sequences of language tokens $x \in \mathcal{X}$ into another. Modern LLMs are composed of two parts, $\mathcal{M} = (E_0^H, M_0)$. The first part is *an embedding layer*, $E_0 : \mathcal{X} \mapsto \mathcal{Z}$ (not to be confused with the domain embeddings $E_D$), which maps tokens $x \in \mathcal{X}$ to their respective *token embedding* representations $z \in \mathcal{Z}$ (as distinct from the domain embedding). The second part is the *the dense model*, $M_0 : \mathcal{Z}^H \mapsto \mathcal{X}^H$, which maps a sequence of token embeddings to a sequence of tokens [2] (e.g., transformer, Vaswani et al. (2017)).

We now define the *Embedding Language Model (ELM)*, a framework which allows one to train a model using textual prompts mixed with continuous domain embedding vectors (see Fig. 2 for an illustration). ELM incorporates domain embedding vectors using *adapter layers* to interface with a text-only LLM. We note that we are not learning $E_D$ itself, but rather wish to interpret it. In other words, given an embedding vector $E_D(v)$, we wish to train an LLM to meaningfully engage in discourse about the entity represented by $E_D(v)$ – even for a hypothetical entity $v$.

**Problem Formulation.** We assume access to a domain embedding space $(\mathcal{W}, d)$, and to pairs $(v, E_D(v))$ for training as explained below. Note that we do not assume access to the mapping $E_D$ (except for empirical evaluation). Furthermore, our technique is agnostic to the type and source of $(\mathcal{W}, d)$. Finally, we assume access to a pretrained text-only LLM $\mathcal{M}$.

To interpret embedding vectors, we require some semantic information about the entities to which they correspond. To this end, let $\mathcal{T}$ be a set of *tasks*, where each task $t \in \mathcal{T}$ is specified by some mapping from joint input sequences $s \in (\mathcal{X} \cup \mathcal{W})^H$ to distributions over output sequences $o \in \mathcal{X}^H$.[3] Intuitively, a task $t$ captures some form of semantic information about the entity represented by $w \in \mathcal{W}$, e.g., the plot of a movie, the preferences of a user, etc. The set of tasks $\mathcal{T}$ should be diverse enough to allow the ELM to extract semantic information from $\mathcal{W}$ to support interpretation, and

---

[2]In practice, the dense model outputs logits over tokens, which are sampled by a decoding procedure.

[3]More generally, we can define output sequences in $(\mathcal{X} \cup \mathcal{W})^H$ as well; we leave this for future work.

ideally support generalization to related tasks. For any $t \in \mathcal{T}$, we assume access to a set of training instances $(s, o)$ drawn from some distribution $P_t$ ($t$ itself may be drawn from $P_{\mathcal{T}}$). For example, a task to compare two movies might use token/domain-embedding input sequences as shown in Fig. 2, and output a textual comparison of the two movies represented by the input embeddings. In general, this does not require knowledge of the item $v$ which corresponds to the input embedding vector $E_D(v)$, but in practice, one will use $v$ to form target output training sequences.

**Architecture.** To account for the embedding space $\mathcal{W}$, we augment the pretrained $\mathcal{M} = (E_0^H, M_0)$ with an adapter model $E_A : \mathcal{W} \mapsto \mathcal{Z}$ to create a new language model $\mathcal{M}_{\mathrm{ELM}} = ((E_0 \times E_A)^H, M_0)$. Here, $E_0$ is the usual embedding layer, which maps tokens representing textual inputs to $\mathcal{Z}$, whereas $E_A$ maps embeddings from the latent metric space $(\mathcal{W}, d)$ to $\mathcal{Z}$. Importantly, $E_0$ and $E_A$ map tokens and embeddings (respectively) to a common space[4] (see Fig. 2).

**Training Procedure.** Given a downstream loss function $\mathcal{L}$, we can differentially optimize the model $\mathcal{M}_{\mathrm{ELM}}^{\theta}$ with parameters $\theta$ over the tasks in $\mathcal{T}$ by solving $\arg\min_{\theta} \mathbb{E}_{s,o \sim P_t; t \sim P_{\mathcal{T}}}[\mathcal{L}_{\theta}(\mathcal{M}_{\mathrm{ELM}}(s, o))]$. Aside from maximum likelihood-based training, we discuss optimizing $\mathcal{M}_{\mathrm{ELM}}$ using reinforcement learning from AI feedback (Lee et al., 2023) in Appendix H.

Training continuous prompts poses challenges due to constraints over the pretrained token embeddings, which may result in convergence to local minima (Liu et al., 2021). Intuitively, as the pretrained embedding layer $E_0$ readily maps to language space, the newly initialized embedding layer $E_A$ may require numerous updates to map to a similar space. To overcome this, we divide training into two stages. In the first stage, we train the adapter $E_A$ on tasks in $\mathcal{T}$ by keeping all other parameters ($E_0$, $M_0$) frozen. Since $M_0$ is pretrained, the learned first-stage mapping from $\mathcal{W}$ to $\mathcal{Z}$ improves convergence in the next stage. In the second stage, we fine-tune the full model by training all parameters ($E_0$, $M_0$, $E_A$). To increase efficiency, we can update a smaller number of parameters in alternating fashion (Hu et al., 2021). We found that two-stage training is essential for convergence of $\mathcal{M}_{\mathrm{ELM}}$ (see Appendix D for further discussion of the two-stage training procedure).

## 3 EXPERIMENT DESIGN

In this section, we describe the datasets, tasks, and evaluation methods used to train and validate ELM. We use the MovieLens 25M dataset (Harper & Konstan, 2015), which we enrich with textual descriptions by generating a large corpus of text using a PaLM 2-L (Unicorn) LLM (Google et al., 2023). We use two different forms of embeddings: *behavioral embeddings* and *semantic embeddings*, which we describe below. We also adopt various evaluation techniques for assessing the quality of ELM's outputs on test data, including qualitative human evaluations and specific consistency metrics.

### 3.1 DATA

The MovieLens 25M dataset contains 25 million ratings (1 to 5) of 62,423 movies and 162,541 users.

**Domain Embeddings.** We create embeddings for both users and movies in the MovieLens dataset. We consider two types of embeddings. The first are *behavioral embeddings*, trained based solely on user ratings of movies. More generally such models can reflect any behavioral interaction between users and items, but use no direct semantic information. To train behavioral embeddings, we use *matrix factorization (MF)* computed using *weighted alternating least squares (WALS)* (Hu et al., 2008), such that the dot product $\hat{r} = \langle w_u^b, w_m^b \rangle$ of the embeddings of user $u$ and movie $m$ is predictive of user $u$'s rating for movie $m$.[5] We train MF using movies that have at least five ratings.

*Semantic embeddings* are the second type, and are generated using textual descriptions of movies. Specifically, we use a pretrained dual-encoder language model (DLM) similar to *Sentence-T5* (Ni et al., 2022a) and *generalizable T5-based dense retrievers* (Ni et al., 2022b). More specifically, we concatenate plot descriptions and reviews for each movie, and input these to the DLM. We then average the resulting output vectors to generate the semantic embeddings. We denote movie $m$'s semantic embedding by $w_m^s$.

---

[4]More expressive adapters that map vectors in $\mathcal{W}$ to length $\ell \geq 1$ sequences are left for future work.

[5]Other CF methods could be used, e.g., dual encoders (Yi et al., 2019; Yang et al., 2020), but our approach is agnostic to the precise method used to generate behavioral or semantic embeddings.

**Training Data and Tasks.** We test two versions of ELM, one that interprets semantic embeddings of movies, and a second which interprets behavioral embeddings of users. For the first model, we use a variety of tasks to test the model's ability to extract reasonable semantic information from the domain embeddings. More specifically, we construct 24 movie-focused tasks using a pretrained PaLM 2-L model. For each task, we generate training data by prompting the PaLM 2-L with a movie's title and additional task-specific information (e.g., writing review, listing characteristics, comparing movies, see details below). We then use PaLM 2-L's generated output as training targets for ELM. As for training inputs, we provide ELM with the same prompts used in PaLM 2-L, where the title of the movie $m$ is replaced by its semantic embedding $w_m^s$. We emphasize that ELM does not receive any information about the movie (including its title), apart from its semantic embedding vector.

Our 24 movie tasks include single movie semantic tasks, such as describing a movie plot or summarizing a movie; single movie subjective tasks, such as writing positive or negative reviews for a movie; and movie pair subjective tasks, such as comparing characteristics of movies. Appendix E provides a complete description of all 24 tasks, including the prompts used, and sample outputs.

Finally, we train an additional model to interpret user behavioral embeddings, by generating *user preference profiles*. For this task, we generate a textual summary of a user's preferences as follows: (a) we sample five positively rated and five negatively rated movies for each user in MovieLens 25M; (b) we then prompt PaLM 2-L to describe (in ten bullet points) characteristics of a person who likes the first five movies, but dislikes the second five. This resulting summary is used as training output for ELM on the *user-profile generation task*. The resulting task is to generate such a user profile from a user's behavioral embedding $w_u^b$ (i.e., interpret the domain embedding). Similar to the movie tasks, no other information about the user, apart from its embedding, is provided to ELM. We refer the reader to Appendix E for more details and examples.

## 3.2 EVALUATION METRICS

We employ several forms of evaluation for ELM. First, given the inherent subjectivity of some tasks (e.g., movie reviews), human raters play a key role in gauging output quality. We ask raters to assess ELM's output w.r.t. its consistency with a movie's plot, its linguistic coherence, and overall task quality. Each output is rated on a scale from 0 (completely irrelevant/incoherent) to 1 (highly relevant/coherent). The average rater's score provides a holistic assessment of ELM's performance.

Second, since our goal is interpretation of embeddings spaces, and should allow the generation of descriptions of domain embedding vectors for which no entity exists in the domain vocabulary (e.g., hypothetical movies), we evaluate the *consistency* of ELM's generated text with the underlying domain embedding, as well as ground-truth data used to train the domain embedding. To achieve this, we introduce two consistency metrics; namely, semantic and behavioral consistency.

*Semantic consistency* (SC) compares the semantic embedding of generated text with the original embedding that produced that text. In our movie tasks, this metric is computed by *re-embedding* ELM's output into the same semantic space used to embed the original movies. Formally, we define the semantic consistency of an output $o$ generated by an embedding $w^s \in \mathcal{W}$ by

$$\text{SC}(o, w^s) = d(P(o), w^s),$$

where $P(o)$ projects the output to the domain embedding space $\mathcal{W}$ and $d$ is the distance metric in $\mathcal{W}$ (or alternatively, a similarity score); see Appendix B, Fig. 6 for an illustration.[6] The utility of SC is grounded in the expectation that if the generated output $o$ holds true to its source $w^s$, then its embedded representation should not deviate substantially from its generating embedding vector. Hence, high SC suggests ELM has well-aligned domain embedding vectors with their (task-specific) descriptions. Importantly, this approach is not tied to examples in the data, providing the flexibility to measure the semantic consistency of *hypothetical entities*.

A second form of consistency, which we term *behavioral consistency* (BC), measures the ability to use ELM's output text to make *good behavioral predictions*. For example, in our user-profile task, we can test the ability of ELM's generated user profile to predict the movie preferences of a user corresponding to the input behavioral embedding. This in turn would suggest the extent to which the user profile captures the information implicit in that domain vector (i.e., the latent user preferences).

---

[6]In our experiments we use the same DLM to serve as $P$, i.e., to re-embed output text into semantic space.

Table 2: Test Scores for 24 movie tasks. We use cosine similarity as our metric for SC, and Spearman rank correlation coeff. for BC. For human evaluation, we show normalized scores between 0 (strongly disagree) and 1 (strongly agree). Table is divided into two sections: single movie tasks and two movie tasks. Tasks with nn abbreviation use nearest neighbor movies in semantic embedding space. A comparison of semantic and behavioral consistency metrics is demonstrated in Fig. 4.

| Task (Appendix E) | Consistency Metrics | | Human Evaluation (100 raters) | | |
|---|---|---|---|---|---|
| | Semantic (Cosine) | Behavioral (Rank Corr.) | Consistency with Plot | Language Comprehensiveness | Task Quality |
| summary | 0.91 | 0.94 | 0.87 | 0.87 | 0.85 |
| positive review | 0.94 | 0.81 | 0.9 | 0.9 | 0.89 |
| negative review | 0.93 | 0.79 | 0.93 | 0.96 | 0.96 |
| neutral review | 0.92 | 0.76 | 1 | 1 | 0.56 |
| five pos char. | 0.91 | 0.8 | 0.84 | 0.92 | 0.95 |
| five neg char. | 0.89 | 0.8 | 0.81 | 0.92 | 0.89 |
| long description | 0.91 | 0.8 | 0.89 | 0.86 | 0.89 |
| funnier | 0.88 | 0.81 | 0.75 | 0.78 | 0.68 |
| sadder | 0.91 | 0.82 | 0.59 | 0.83 | 0.81 |
| scarier | 0.9 | 0.84 | 0.69 | 0.74 | 0.95 |
| improve | 0.91 | 0.82 | 0.85 | 0.89 | 0.87 |
| movie to viewer | 0.89 | 0.8 | 0.89 | 0.83 | 0.86 |
| pitch | 0.96 | 0.8 | 0.96 | 0.95 | 0.94 |
| criticize | 0.89 | 0.77 | 0.88 | 0.7 | 0.79 |
| convince1 | 0.92 | 0.96 | 0.98 | 0.93 | 0.77 |
| convince2 | 0.9 | 0.88 | 0.89 | 0.86 | 0.87 |
| convince3 | 0.88 | 0.86 | 0.7 | 0.67 | 0.74 |
| dissuade1 | 0.88 | 0.85 | 0.88 | 0.89 | 0.87 |
| dissuade2 | 0.88 | 0.87 | 0.79 | 0.8 | 0.79 |
| similarities | 0.86 | 0.83 | 0.37 | 0.4 | 0.38 |
| interpolation | 0.89 | 0.8 | 0.85 | 0.82 | 0.84 |
| why like nn | 0.83 | 0.48 | 0.84 | 0.84 | 0.91 |
| diff than nn | 0.82 | 0.48 | 0.61 | 0.79 | 0.64 |
| common with nn | 0.77 | 0.46 | 0.91 | 0.86 | 0.93 |

To achieve this, we assume access to some "off-the-shelf" language-based retriever/ranker $\rho$ which, given a user profile, can rank a set of candidate movies. Then, given a fixed set of target movies, $\mathcal{I}$, and letting $o_u$ be the ELM-generated user profile given domain embedding $w_u^b$, we define

$$\text{BC}(o_u, u) = \text{RankingScore}(R_u, \rho(o_u)),$$

where $\rho(o_u)$ are the ranks of the movies in $\mathcal{I}$ for user $u$ provided by the language-based ranker, and $R_u = \{r(u, m) : m \in \mathcal{I}\}$ are the (MovieLens) ground-truth ratings for $\mathcal{I}$ provided by $u$ (converted to rankings); see Appendix B, Fig. 7 for an illustration.[7] Here, *RankingScore* is any rank-comparison metric which compares $\rho(o_u)$ to the ground-truth ranking $R_u$. Examples include *NDCG* (Burges et al., 2005) and the *Spearman rank-correlation coefficient*. Finally, we measure behavioral consistency of movies in a fully analogous fashion (see Appendix B for a formal definition).

## 4 EMPIRICAL ANALYSIS

We evaluate ELM on our 24 movie tasks and the user profile task, described in Sec. 3.1, using human raters, and semantic and behavioral consistency to measure performance. We analyze test results on both real entities (i.e., held-out movies) and hypothetical embedding vectors not in the data, to test ELM's ability to interpolate between movies or users, as well as extrapolate movie or user attributes.

We train two models: one (joint) model for the 24 movie tasks, and another for user profiles. For the movie tasks, we use 1000 randomly sampled examples for test and the rest for training. For the user profile task we use an 80/20 training/test split. Both models are fine-tuned using a pretrained

---

[7]In our experiments we use the dual-encoder DLM to serve as $\rho$, by embedding test movies and user profiles and using their similarity for scoring/ranking.

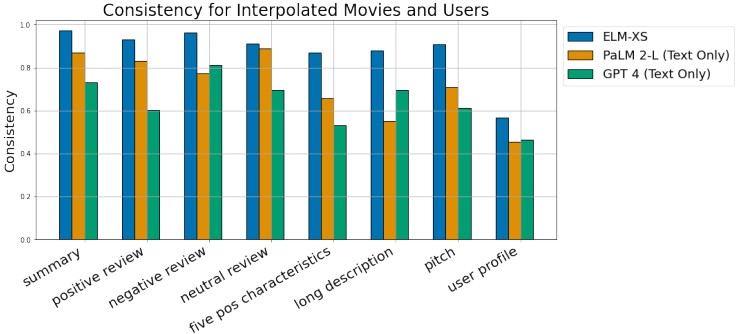

Figure 3: Results show consistency results for movie and user interpolations using ELM, compared to one-shot decoded results of text only LLMs. We show semantic consistency (cosine) scores for movies and behavioral consistency (NDCG) scores for user profiles.

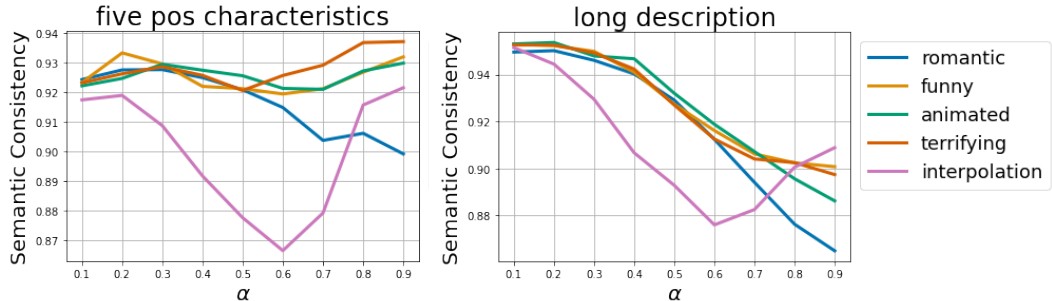

Figure 4: Plot depicts SC scores (using cosine similarity) for interpolations of movies with the movie "Forrest Gump", as well as extrapolation of CAV attributes in semantic space.

**PaLM 2-XS (Otter).** We use a two-layer MLP for the domain-embedding adapter layer $E_A$, and two-stage training, as described in Sec. 2. Particularly, for movie tasks, we run the first stage of training for 20k iterations, and then the second stage for another 300k iterations, using a batch size of 32 (i.e., roughly seven epochs). We use a similar procedure for user profiles. We found training in two stages to be essential for convergence of ELM (see Appendix D for further details).

**Test-item Evaluation.** Table 2 shows consistency (SC and BC) and human rater results for all movie tasks on the test set. For each task, we ask 100 human raters[8] to rate the consistency and comprehensiveness of our results w.r.t. the ground-truth data. We also ask qualitative task-specific questions (Appendix C describes rater instructions in detail). Rater evaluations show that ELM is able to generalize well across most tasks. Moreover ELM maintains high semantic consistency across all tasks and good behavioral consistency for most. We note that performance tends to be worse on two-movie tasks, due to the additional challenge of "conversing" about similarities and differences between two distinct embedding representations. We emphasize the importance of our consistency metrics, as they reflect how well ELM adapts to the domain embedding space $\mathcal{W}$.

**Communicating Novel Entities.** We next test ELM's capability to meaningfully extrapolate from existing movies or users to interpret "gaps" in embedding space (e.g., embedding vectors of hypothetical movies for which we predict a large audience). Note that such hypothetical vectors have no ground-truth data for evaluation; nor is it clear how to evaluate hypothetical embeddings using text-only LLMs (e.g., Google et al. (2023); OpenAI (2023)). For instance, there is no obvious way to instruct a text-only LLM to describe an arbitrary vector in $\mathcal{W}$. Even if we were to ask a text-only LLM to interpolate two existing entities (e.g., movies), the result will not necessarily be consistent with the domain embedding space we wish to interpret, nor can one in general direct a text-only LLM to an arbitrary point using interpolation between a small number of existing entities.

---

[8]Raters were paid contractors. They received their standard contracted wage, which is above the living wage in their country of employment.

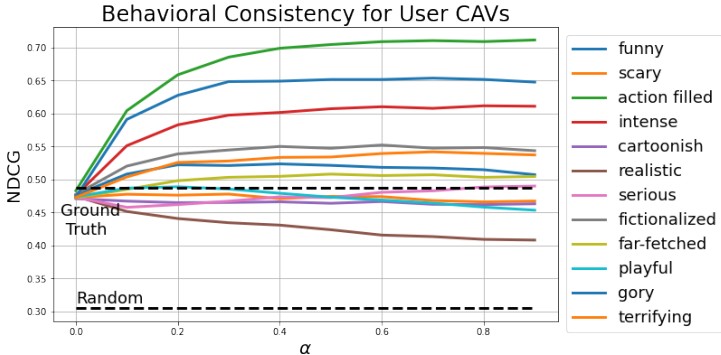

Figure 5: Plot depicts BC scores (using NDCG with the predicted CF model ratings) for user profile decoded outputs, where user embeddings are extrapolated in different CAV directions. Dotted lines show two baselines (for $\alpha = 0$): the random baseline, which rates movies randomly, and the ground-truth baseline, which uses the ground-truth user profile data to compute BC (i.e., BC of the ground-truth training data).

Behavioral embeddings of, say, users are even more challenging for text-only LLMs. This puts additional emphasis on the need to evaluate such LLMs using embedding consistency metrics like SC and BC. That said, to ground our results, we do not consider arbitrary hypothetical points in embedding space. Instead, we evaluate our model using interpolation of *existing entities*, as well shifts of such entities in specific directions (see **Generalizing CAVs** below). Nevertheless, ELM's ability to interpret hypothetical embedding vectors is not constrained to these choices.

We first assess consistency metrics on interpolations of two movies or users. Fig. 3 compares the performance of ELM with state-of-the-art text-only LLMs; namely, PaLM 2-L and GPT 4. For movies, we linearly interpolate (with mixture coefficient $\alpha = 0.5$) embedding vectors of movies in the training set with the classic movie "Forrest Gump (1994)", and provide the *interpolated embedding* to ELM. For users, we interpolate random pairs of user embedding vectors from the training set. For a fairer comparison, we provide the text-only LLMs with a ground-truth examples of each task's output, and prompt them to generate the task output. Particularly for users, we provide the text-only LLMs with ground-truth user profiles, and instruct them to interpolate between them (see Appendix F for the precise prompts used). We measure semantic consistency for movie tasks and behavioral consistency for the user profile task. We found that text-only LLMs are significantly less consistent than ELM. This is not surprising, as describing a domain embedding space manifold to a text-only LLM is especially challenging. Moreover, ELM is able to generate high-quality, consistent responses to our tasks. We provide further qualitative results in Appendix G.

We next test ELM's consistency as we vary the degree of interpolation between movies, by varying the mixture weight $\alpha$. Fig. 4 shows semantic consistency (magenta line) for two of our movie tasks. We find that outputs are highly consistent across all degrees of interpolation values, with a tendency for lower consistency scores for embeddings that are farther away from the two movies. This result is unsurprising, as we expect performance to degrade as it moves away from "real" data points. Also note these are linear interpolations, though the true embedding manifold may be highly non-linear.

**Generalizing Concept Activation Vectors (CAVs).** Finally, we test the interpretation of vectors in domain embedding space by extrapolating existing entity embeddings in specific directions. To accomplish this, we train CAVs [9] (Kim et al., 2018) using attribute tags in MovieLens 25M, provided by movie raters. These CAVs can be treated as noisy, "soft" movie attributes, such as 'funny,' 'terrifying,' 'thought-provoking,' etc. The CAV corresponding to a specific attribute provides a *direction* in domain embedding space that represents an increase in that attribute's value. We use these to *extrapolate* movies and users in specific CAV directions (e.g., make an existing movie funnier, or tweak a user profile to prefer more cartoonish movies).

In Fig. 4, we see that semantic consistency of movie extrapolations generally decreased as we extrapolated further in CAV directions. This is expected as extrapolations outside the data manifold should

---

[9]We train CAVs using the procedure of Göpfert et al. (2022) and the dataset of Balog et al. (2021), to which we refer for details. Specifically, we use linear, objective rather than non-linear and/or subjective CAVs.

induce higher error. Similarly, Fig. 5 shows behavioral consistency results for CAV extrapolations of user profiles. These results show attribute extrapolations to be behaviorally consistent for different $\alpha$ values. Perhaps surprisingly, user profiles show *greater* behavior consistency as we increase the degree to which they move in a CAV direction. We hypothesize this is due to user profiles becoming "more predictable" in our CF model by making user preferences more extreme. For instance, as we make a user profile gravitate to funnier movies, preferences tend to degenerate with a near-exclusive focus on funny movies (thus rendering rating prediction – and especially relative rating prediction – more accurate). This in turn increases BC. We provide further qualitative results for interpolations and CAV-based extrapolations in Fig. 1. We refer to Appendix G for more qualitative results.

## 5 RELATED WORK

We review a selection of related work. The use of embeddings in NLP is well-established, with seminal work like *Word2Vec* (Mikolov et al., 2013) and *GloVe* (Pennington et al., 2014) using dense vectors to represent words in a way that captures semantics. These approaches serve as the foundation for many models in NLP, e.g., contextual (Bengio et al., 2000; Collobert & Weston, 2008; Radford et al., 2018) and deep-averaging networks (Iyyer et al., 2015). Bowman et al. (2015) show that interpolating sentences in embedding space can generate novel sentences that smoothly blend the "ideas" in the original sentences. Such NLP models provide some "immediate" interpretability.

Our ELM approach supports interpretation of *arbitrary* embedding spaces using natural language. In this sense, ELM can be seen as learning an encoder-decoder model (Raffel et al., 2020), for which the encoder adapts the domain-embedding to the token-embedding space, and the LLM acts as decoder from the "semantics" latent in the domain embedding to natrual language. As such, our work is closely related to research on translation (Vaswani et al., 2017) and multi-modality (Cho et al., 2021; Yu et al., 2022) that rely on encoder-decoder architectures. A key difference with ELM lies in the fact that encoder inputs (domain embeddings) are not igenerally interpretable; by contrast, in translation and multi-modal tasks, labeled data can be explicitly acquired from humans.

The use of continuous vectors as prompts for language models can provide flexibility in controlling outputs, improving performance on downstream tasks. Examples include: prefix tuning (Li & Liang, 2021; Lester et al., 2021; Tsimpoukelli et al., 2021), which prepends a sequence of task-specific vectors to the input, while keeping LLM parameters frozen; and hard-soft prompt hybrid tuning (Liu et al., 2021; Han et al., 2022), which inserts tunable embeddings into a hard prompt template. These methods primarily guide model behavior for downstream tasks, rather than *interpreting* embeddings, especially those derived from different data sources (which we want to understand, not create).

CAVs (Kim et al., 2018) and visual techniques such as t-SNE (Van der Maaten & Hinton, 2008) or UMAP (McInnes et al., 2018) have been used for visualizing and interpreting embeddings. Nevertheless, these methods often provide static insights or visual interpretations, and are often hard or even impossible to capture in a low dimension (Chari & Pachter, 2023; Seshadhri et al., 2020). ELM, by contrast, seeks to enable dynamic, language-based exploration of the embedding space, "visualizing" it by narrating embeddings with language. Other work has focused on interpretability analysis and toolkits for language and sequence models (Belinkov et al., 2020; Sarti et al., 2023; Bills et al., 2023; Räuker et al., 2023), but these methods are largely orthogonal to our work.

## 6 CONCLUSION

We have presented ELM, a novel language-model framework for interpreting domain-embedding spaces. We assessed ELM's capabilities on a range of movie tasks and a user-profile task, benchmarking with both human evaluations as well as two novel quantitative metrics; namely, semantic and behavioral consistency. Our results show that ELM generalizes well to embedding vectors in a test set, and aligns well with human-rater expectations. We also find that ELM is adept at handling the nuanced challenge of describing novel entities, with better semantic and behavioral in tasks that interpolate between entities, compared to state-of-the-art text-only LLMs. We also demonstrated ELM's proficiency in generalizing CAVs, underscoring its ability to meaningfully manipulate and interpret domain attributes of movies and user profiles. Taken together, our results suggest that ELM offers a powerful, flexible mechanism for understanding, navigating and manipulating complex embedding representations.

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

Table 3: A short description of each of the Amazon tasks.

| | |
|---|---|
| description | Description of product. |
| positive review | A positive review of the product (rating 5). |
| negative review | A negative review of the product (rating 1-3) |
| neutral review | A neutral review of the product (rating 4) |
| endorsement | An endorsement of the product. |
| user profile | Ten bullet points describing a user based on positive and negative reviews. |

Table 4: Test Scores for Amazon tasks.

| Task (Appendix E) | Consistency Metrics | | Human Evaluation (100 raters) | | |
|---|---|---|---|---|---|
| | Semantic (Cosine) | Behavioral (Rank Corr.) | Consistency with Item / User | Language Comprehensiveness | Task Quality |
| description | 0.93 | 0.92 | 0.81 | 0.88 | 0.75 |
| positive review | 0.95 | 0.89 | 0.76 | 0.79 | 0.93 |
| negative review | 0.91 | 0.94 | 0.77 | 0.76 | 0.71 |
| neutral review | 0.96 | 0.82 | 0.8 | 0.79 | 0.7 |
| endorsement | 0.9 | 0.81 | 0.91 | 0.93 | 0.94 |
| user profile | 0.89 | 0.87 | 0.84 | 0.77 | 0.75 |

## A ADDITIONAL EXPERIMENTS ON THE AMAZON PRODUCT DATASET

We complement our experiments and demonstrate ELM on the public Amazon dataset (McAuley et al., 2015), which consists of 9.35M items with textual descriptions, 20.9M users, 233.1M reviews, and 82.83M ratings. We focus specifically on a subset of this dataset in the category "Clothing, Shoes and Jewelry", which consists of 5.7M reviews for 1.5M products.

**Domain Embeddings.** Similar to our setting in the Movielens dataset, we create both semantic and behavioral embeddings. We train behavioral embeddings on users' rating data using the same method described in Sec. 3.1. For semantic embeddings, we use DLM (see Sec. 3.1) to encode a concatenation of the item title, item description, item categories, and item features (which include item tags).

**Training Data and Tasks.** We test ELM's ability to interpret semantic and behavioral embeddings on a set of six tasks: item description, positive review, negative review, neutral review, item endorsements, and user profile. Item descriptions are taken directly from the Amazon dataset. For review data, we split the Amazon reviews w.r.t. to positive, negative, and neutral reviews. To ensure the datasets are of comparable size, we use scores of 1-3 to represent negative reviews, a score of 4 to represent neutral reviews, and a score of 5 to represent positive reviews. To create the item endorsement task we prompt a PaLM 2-L with the item title and description and ask it to create an endorsement for that product. Finally, for user profiles, we use a similar procedure as in Movielens. Specifically, we describe a user using positive and negative reviews they have posted, and ask a PaLM 2-L to describe a user with such reviews in ten bullet points. We note that, similar to the Movielens tasks, the Amazon training data does not consist of any information about items or users except for the behavioral or semantic embedding information. Table 3 summarizes the set of tasks we use.

**Results.** Table 4 shows consistency (SC and BC) and human rater results for the Amazon dataset tasks on a test set of 1000 examples (randomly selected). For each task, we ask 100 human raters to rate various aspects of the outputted result, such as its consistency with the true item, comprehensiveness of the language, and other task-specific quality evaluations. Simliar to the Movielens experiments, our results show that ELM is able to generalize well across all tasks. Moreover ELM maintains high semantic and behavioral consistency across all tasks.

We conduct further detailed human evaluation studies on the Amazon tasks. For this evaluation we ask human raters to evaluate both ground-truth data as well as results generated by our model. For example, we ask human raters to evaluate the quality of a review in the real data w.r.t. the ground-truth item description. This evaluation enables us to obtain a baseline for the model's scores. Below we

Table 5: Human evaluation scores on generated data based on form questions (Q). Table shows average scores of 100 human raters.

| Task | Q1 | Q2 | Q3 | Q4 | Q5 | Q6 | Q7 | Q8 | Q9 | Q10 |
|---|---|---|---|---|---|---|---|---|---|---|
| description | 0.77 | 0.7 | 0.86 | 0.71 | 0.73 | 0.7 | 0.81 | 0.9 | 0.9 | 0.9 |

| Task | Q1 | Q2 | Q3 | Q4 |
|---|---|---|---|---|
| positive review | 0.91 | 0.92 | 0.92 | 0.98 |
| negative review | 0.75 | 0.72 | 0.71 | 0.75 |
| neutral review | 0.85 | 0.8 | 0.79 | 0.7 |

| Task | Q1 | Q2 | Q3 |
|---|---|---|---|
| endorsement | 0.91 | 0.93 | 0.93 |

Table 6: Human evaluation scores on ground-truth data based on form questions (Q). These evaluations are on test data, not generated output. Table shows average scores of 100 human raters.

| Task | Q1 | Q2 | Q3 | Q4 | Q5 | Q6 | Q7 | Q8 | Q9 | Q10 |
|---|---|---|---|---|---|---|---|---|---|---|
| description | 0.68 | 0.64 | 0.85 | 0.83 | 0.82 | 0.83 | 0.84 | 0.74 | 0.73 | 0.73 |

| Task | Q1 | Q2 | Q3 | Q4 |
|---|---|---|---|---|
| positive review | 0.85 | 0.76 | 0.79 | 0.92 |
| negative review | 0.75 | 0.77 | 0.74 | 0.71 |
| neutral review | 0.84 | 0.79 | 0.81 | 0.7 |

present examples of forms that were presented to the raters, with task-specific questions. In Tables 5 and 6 we show the average human rater answers to these questions. We find that our generated results correctly correspond to test items, and perhaps surprisingly, often surpasses the original ground-truth data quality, as viewed by the human raters.

---

**Item Description Task (Rater Form Example)**

**BELOW ARE TWO DESCRIPTIONS OF AN AMAZON PRODUCT TITLED "TIMEX WOMEN'S T2P0902M LEOPARD PATTERNED LEATHER STRAP WATCH"**

**FIRST DESCRIPTION:**

Mens Hipster Wallet with Genuine Leather. Perfect wallet to keep everything organized. With a slot for cash and ample credit card space, this item is great! It even has an easily accessible ID window! 8+ Card slots built into the wallet help to keep this item slim.

**SECOND DESCRIPTION:**

This slim mens wallet keeps essentials organized in a convenient wallet. With multiple card slots, ID window and cash pocket, this wallet keeps everything together. Keep up with your essentials with this wallet.. Please note: This item is shipped via USPS First Class Package and usually takes 5-7 business days to arrive. If you would like the item faster, please choose a different shipping method such as UPS or FedEx Express..

On scale of 1-5, how much do you agree with the following:
1 = Strongly Disagree
5 = Strongly Agree

**Question 1:** Both descriptions are equivalent.
**Question 2:** The descriptions are similar.
**Question 3:** Both descriptions could correspond to the same product.
**Question 4:** The first description is of high quality.
**Question 5:** The first description is coherent.
**Question 6:** The first description is comprehensive.
**Question 7:** The first description is relevant to the item title.
**Question 8:** The second description is coherent.
**Question 9:** The second description is comprehensive.
**Question 10:** The second description is relevant to the item title.

---

**Positive Review Task (Rater Form Example)**

**BELOW YOU WILL BE PRESENTED WITH A DESCRIPTION FOR AN AMAZON PRODUCT TITLED "HANES WOMEN'S 10 PAIRS CUSHIONED SOLE LOW CUT SOCKS SZ: 9-11 FITS SHOE 5-9". YOU WILL THEN BE ASKED TO RATE A REVIEW OF THIS PRODUCT.**

**DESCRIPTION OF PRODUCT:**

The road can be rough. Wearing these socks help make it a little smoother. Provides a better fit with spandex.Soft, breathable cotton knit helps you keep your cool.Plush cushioned foot bottom soothes your every step.Reinforced grey heel and toe resist wear & tear.

**BELOW IS A REVIEW FOR THE PRODUCT ABOVE.**

I am a 7.5 to 8 medium weight woman with a high instep. These socks fit perfectly and are not too tight around the top. They stay up well and are comfortable to wear all day. The colors and patterns are fun. I would definitely buy these again. I did not find them to be that slippery on floors. Maybe I was wearing them right after they arrived and they need a little breaking in. I do think they are better than most athletic socks in that they are more like a thin mid weight sweater than a stocking. They are good to wear with jeans or leggings. I was happy to find that they are good quality and not made in China. The label says they are Shrink To Fit which means they use elastic in the toe and heel. That should help them keep their shape. I will follow up with a comment after I have worn them for a while. UPDATE: I have worn these socks for 3 months now. I wear them to work 5 days a week. I wear them with dress shoes most of the time. They have held up well with little wear on the heel.

On scale of 1-5, how much do you agree with the following:
1 = Strongly Disagree
5 = Strongly Agree

**Question 1:** It is reasonable to assume this review would appear for this product.
**Question 2:** The product review is consistent with the product's description.
**Question 3:** The product review is coherent.
**Question 4:** The product review is positive.

---

**Negative Review Task (Rater Form Example)**

**BELOW YOU WILL BE PRESENTED WITH A DESCRIPTION FOR AN AMAZON PRODUCT TITLED "PT261 WOMEN'S ATHLETIC PERFORMANCE LOOSE FORM FIT RACER BACK FITNESS TOP". YOU WILL THEN BE ASKED TO RATE A REVIEW OF THIS PRODUCT.**

**DESCRIPTION OF PRODUCT:**

The Game Gear Women's Performance Tech Racer Back Shirt is a form fit moisture wicking athletic top. This sleek design will provide ultimate comfort and unrestricted shoulder and arm mobility. Great for the gym, outdoor activities, or any sport.

**BELOW IS A REVIEW FOR THE PRODUCT ABOVE.**

Waaaaay too big. I would order a size down. I'm 5'4", 135lbs and ordered the small/medium and it was like a sack. Great as a night gown or lounging piece. I do love the material. Soft and comfortable. I would order a medium if you are normal weight or smaller. I ended up giving mine to my 10 year old daughter and she loves it. I would order a smaller size for sure. I did enjoy the time I had with it. :-) Peace out. I hope this review was helpful. I don't usually write reviews, but I was really disappointed with the size of this shirt. It's a shame because it's a great material. Peace out. I hope this review was helpful. :-) Peace out. I don't usually write reviews, but I was really disappointed with the size of this shirt.

On scale of 1-5, how much do you agree with the following:
1 = Strongly Disagree
5 = Strongly Agree

**Question 1:** It is reasonable to assume this review would appear for this product.
**Question 2:** The product review is consistent with the product's description.
**Question 3:** The product review is coherent.
**Question 4:** The product review is negative.

---

**Neutral Review Task (Rater Form Example)**

**BELOW YOU WILL BE PRESENTED WITH A DESCRIPTION FOR AN AMAZON PRODUCT TITLED "JERRISAP-PAREL NEW CINDERELLA DRESS PRINCESS COSTUME BUTTERFLY GIRL". YOU WILL THEN BE ASKED TO RATE A REVIEW OF THIS PRODUCT.**

**DESCRIPTION OF PRODUCT:**

JerrisApparel New Cinderella Dress Ella Princess Costume Butterfly Girl For 2-9 Years Detailed Size Information(in Inches) Notice: There do exist 1-2 inches differences because of different measuring methods. Please check the size info carefully. Thank you for your understanding. 3 Years: recommended height:39-43.3 chest/bust measurement:19.5-24.5 waist measurement:18-22 dress length:32 4 Years: recommended height:43.3-47.2 chest/bust measurement:20.5-25.5 waist measurement:19-23 dress length:34 5 Years: recommended height:47.2-51 chest/bust measurement:21.5-26.5 waist measurement:20-24 dress length:35 6 Years: recommended height:51-55 chest/bust measurement:22.5-27.5 waist measurement:21-25 dress length:36 7 Years: recommended height:55-59 chest/bust measurement:23.5-30 waist measurement:22-26 dress length:38 8 Years: recommended height:59-62 chest/bust measurement:27-31.5 waist measurement:26-29 dress length:40

**BELOW IS A REVIEW FOR THE PRODUCT ABOVE.**

My daughter is 4 and a half and wears a size 5t. I ordered the 4-6 and it was too big. The skirt was huge. I should have gotten a smaller size but nonetheless, she loved it. I had to return it though because it was too big and i already had a similar dress. So i would definitely recommend this dress but get a smaller size. It is very cute and well made. My daughter loved it. I gave it 4 stars because it is really cute and well made but it was too big. I would have given it 5 stars if it had come in a smaller size.

On scale of 1-5, how much do you agree with the following:
1 = Strongly Disagree
5 = Strongly Agree

**Question 1:** It is reasonable to assume this review would appear for this product.
**Question 2:** The product review is consistent with the product's description.
**Question 3:** The product review is coherent.
**Question 4:** The product review is neither positive nor negative.

---

**Endorsement Task (Rater Form Example)**

**BELOW YOU WILL BE PRESENTED WITH A DESCRIPTION FOR AN AMAZON PRODUCT TITLED "YOUYOUPIFA SILVER PARTIALLY HOLLOW TRANSPARENT DIAL STAINLESS STEEL BAND AUTOMATIC MECHANICAL WATCHES". YOU WILL THEN BE ASKED TO RATE AN ENDORSEMENT OF THIS PRODUCT.**

**DESCRIPTION OF PRODUCT:**

Gender:Unisex Movement Description:Automatic Mechanical Movement Case Size:43*40 mm / 1.69*1.57 inch Case Thickness:14 mm / 0.55 inch Strap Width:18 mm / 0.71 inch Bezel:Stainless steel bezel Dial:Silver partially hollow transparent dial Case:Satin & Polished finish case Case Back:Transparent case back that the movement of the watch can been seen Crystal:High Quality Organic Crystal- Maximum Scratch Resistant Strap:Stainless steel band Folder:Fold-Over-Clasp Weight:100 g Length:18.5 cm Water resistant:Daily water resistant, please don't put it in water

**BELOW IS AN ENDORSEMENT FOR THE PRODUCT ABOVE.**

This is a beautiful, unisex watch with a silver partially hollow transparent dial and a stainless steel band. It is an automatic mechanical watch, which means that it does not require a battery. The watch is also water resistant, so you can wear it in the rain or while washing your hands. The watch is 43*40 mm in size and 14 mm thick. The strap is 18 mm wide. The watch has a stainless steel bezel and a satin and polished finish case. The case back is transparent, so you can see the movement of the watch. The crystal is high quality organic crystal, which is scratch resistant. The strap is made of stainless steel and has a fold-over clasp. The watch weighs 100 g and is 18.5 cm long. This is a great watch for anyone looking for a stylish and durable timepiece.

On scale of 1-5, how much do you agree with the following:
1 = Strongly Disagree
5 = Strongly Agree

**Question 1:** The endorsement is consistent with the product's description.
**Question 2:** The endorsement is coherent.
**Question 3:** The endorsement is trying to convince someone the product is good for them.

---

Below we provide several examples of user profiles in the Amazon dataset, as generated from positive and negative reviews. We used these profiles as training data for ELM to generalize to unseen user embeddings.

## Example 1 of User Profile

Positive review:

Positive review: I travel around the country weekly for business and sometimes need to run down airport concourses to catch my plane; these are the perfect shoes for business attire as well comfort throughout week. I highly recommend them!

Negative review:

I really wish I hadn't thrown all the shipping materials away before trying these on; they would have been returned. Shoe is very heavy, very wide for a wide width and even with inserts, heel slip protectors, etc. I still can't use them. Goodwill has gotten a never worn pair of shoes.

User Profile:

- The user is a business traveler who needs shoes that are both comfortable and stylish.
- The user prefers shoes that are not too heavy or wide.
- The user is willing to pay for high-quality shoes.
- The user is not afraid to try new brands or styles.
- The user is willing to return shoes if they are not satisfied.
- The user is likely to donate shoes that they do not wear.
- The user is likely to buy shoes online.
- The user is likely to read reviews before buying shoes.
- The user is likely to buy shoes from a trusted retailer.

## Example 2 of User Profile

Positive review:

It seems exactly like it was advertised. I was pleased with this product. It has a nice design and the price is good.

Negative review:

The necklace is attractive but it has a poor design. It refuses to hang like it is shown. Instead it turns up-side-down so the under side of it shows and the pretty outside design is inside toward my neck. The two chain attachments should be nearer one another instead of 180 degrees apart. Then it would not turn up-side- down and inside-out. If the attachments were on the top - kind of like Micky Mouse ears - then the pendant would hang like it is supposed to. Please change the points of the chain attachments and I will order it , but I am sending this one back.

User Profile:

- The user likes products that are as advertised.
- The user likes products with a nice design.
- The user likes products that are affordable.
- The user dislikes products that have a poor design.
- The user dislikes products that do not hang properly.
- The user dislikes products that are upside down.
- The user dislikes products that are inside out.
- The user dislikes products that have chain attachments that are 180 degrees apart.
- The user would like products that have chain attachments that are nearer to one another.
- The user would like products that have chain attachments that are on the top.

**Example 3 of User Profile**

Positive review:

The hat fits great! I have a large head, and have a difficult time finding hats that fit. The yarn is soft as opposed to scratchy, and after trying it on and wearing it a while, it doesn't slip off, or look like I shopped in the kiddie beanie section.

Negative review:

The hat fits great! I have a large head, and have a difficult time finding hats that fit. The yarn is soft as opposed to scratchy, and after trying it on and wearing it a while, it doesn't slip off, or look like I shopped in the kiddie beanie section.

User Profile:

- The user is a woman.
- The user has a large head.
- The user has difficulty finding hats that fit.
- The user prefers soft yarn over scratchy yarn.
- The user does not want hats that slip off.
- The user does not want hats that look like they are from the kiddie beanie section.
- The user wears women's size 11/12 or men's size 10 shoes.
- The user does not want to wear socks in the summer.
- The user does not want to mess with getting the right size of shoes.
- The user thinks the shoes look great.

**Example 4 of User Profile**

Positive review:

Purchased for a wedding. Love them!! A shoe I can wear over and over. Great purchase.

Negative review:

When I ordered this dress I thought I was going to get an open, breezy, A-line, asymmetrical dress with a lot of movement. Instead the dress is basically straight with two awkward and uncomfortable flaps sewn to it (one on each side) to give the impression that the dress is open and would have a lot of movement. What this looks like on me is a straight fitting, body hugging t-shirt dress with two fabric wings sewn to each side. Perhaps I misread the description, but this is not how I interpreted the fit. Adding to this unfortunate and unflattering design is that the added panels also serve to add a lot of weight and coupled with the narrowness of the fit, serves to make the dress uncomfortable. I looked to see if I could simply remove the seams from the odd side flaps to open the dress up, but that would serve to ruin the dress. I would say that, for me, it is unwearable and too uncomfortable to even sleep in. I will say that it is cheap enough that I have not returned it because that feels like more hassle than what I originally paid for it. I was very disappointed.

User Profile:

- The user likes shoes that can be worn over and over again.
- The user likes shoes that are comfortable.
- The user likes shoes that are stylish.
- The user likes shoes that are affordable.
- The user likes dresses that are open and breezy.
- The user likes dresses that are A-line and asymmetrical.
- The user likes dresses that have a lot of movement.
- The user likes dresses that are not too tight or body hugging.
- The user likes dresses that are made of lightweight fabrics.

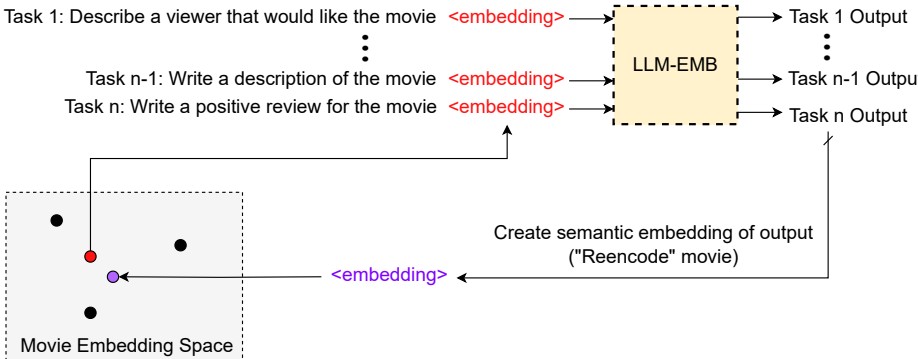

Figure 6: Semantic Consistency (SC). Task output is "re-encoded" back to semantic embedding space. A distance metric is then used to capture the consistency to the original embedding vector that to generate the output.

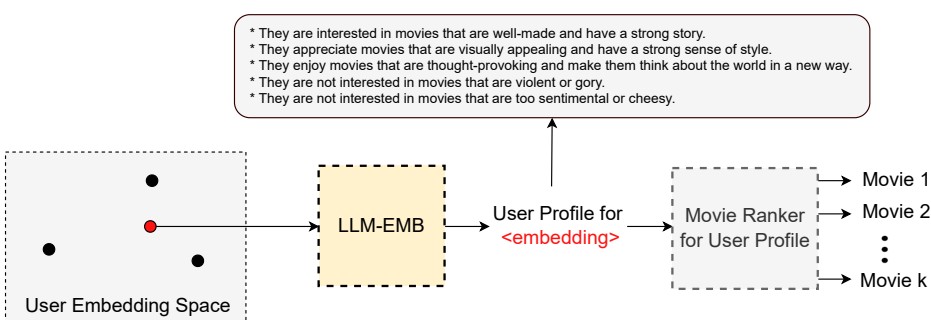

Figure 7: Behavioral Consistency (BC). Diagram depicts BC for the user profile task. Here, a generated user profile is used as input to a Movie Ranker, which ranks movies based on the user profile. The ranks of the movies are then compared to their groundtruth ranks in the MovieLens data, which are carried by the behavioral embedding that was used to generate the user profile.

## B  SEMANTIC AND BEHAVIORAL CONSISTENCY

**Implementation Details of Semantic and Behavioral Consistency.**   While human raters can assess any semantic difference between the ground-truth text and the generated output, we measure the semantic consistency by computing cosine similarity between the semantic embedding of the ground-truth text and the semantic embedding of generated output. For movie tasks, it is more straightforward as the input embedding to ELM is indeed the semantic embedding of the ground-truth text and we also employ the same DLM to *reencode* generated outputs by ELM as shown in Figure 6. For the user profile task, the input embedding is a behavioral one though both the input embedding and the ground-truth user profile come from the same user. Figure 7 describes how we define behavioral consistency for users. The input behavioral embedding is predictive of movie rankings and we want to see if the movie ranker gives consistent rankings for randomly selected movies. We impute zero-star if a selected movie has no rating. The movie ranker ranks movies based on similarity to the generated user profile in the semantic embedding space defined by some DLM.

**Behavioral Consistency for Movies.**   Similar to behavioral consistency for users, as defined in Sec. 3.2, we can define behavioral consistency for movies. Letting $\mathcal{U}$ denote a set of users, for any movie task with generated output $o_m$ of movie $m$, we define behavioral consistency as

$$\text{BC}(o_m, m; \mathcal{U}) = \text{RankingScore}(R_m || \rho(o_m)),$$

where $\rho(o_m)$ are the ranks of the users in $\mathcal{U}$ for movie $m$, provided by a *User Ranker* which uses $o_m$ as its query, and $R_m = \{r(u, m) : u \in \mathcal{U}\}$ are the MovieLens ground-truth ratings for users $\mathcal{U}$ and movie $m$.

Figure 8: Evaluation Template for Running Human Rater Experiment for the "funnier" Task.

## C  RATER EVALUATION

We asked 100 human raters to evaluate quality of all 24 movie tasks, in terms of consistency with plot, language comprehensiveness, and task quality. For each task we randomly generated 40 model utterances with movies not used in training, save each utterance along with the contexts and questions on a Google form shown in Figure 8, and asked a rater to evaluate. Usually we start from ground-truth data of a movie and then model utterance generated for the task. For questions, we always ask *how much do you agree with the following* statement. Below we list questions asked for each task.

1. **long description**
    - The second version plot is consistent with the first version.
    - The second version plot is coherent.
    - The second version plot is high quality like the first version.

2. **summary**
    - The movie summary is consistent with the movie plot.
    - The movie summary is coherent.
    - The movie summary is of high quality.

3. **positive review**
    - The movie review is consistent with the movie plot.
    - The movie review is coherent.
    - The movie review is positive.

4. **negative review**
    - The movie review is consistent with the movie plot.
    - The movie review is coherent.
    - The movie review is negative.

5. **neutral review**
    - The movie review is consistent with the movie plot.
    - The movie review is coherent.
    - The movie review is neutral.

6. **five pos characteristics**

- The characteristics listed are consistent with the movie plot.
- The characteristics listed are coherent.
- The characteristics listed are positive.

7. **five neg characteristics**
    - The characteristics listed are consistent with the movie plot.
    - The characteristics listed are coherent.
    - The characteristics listed are negative.

8. **improve**
    - The improved version plot is consistent with the original plot.
    - The improved version is coherent.
    - Some viewers may find the improved version to be better than the original.

9. **criticize**
    - The criticism is consistent with the movie's plot.
    - The criticism is coherent.
    - The criticism is of good quality.

10. **pitch**
    - The pitch for the movie is consistent with the movie's plot.
    - The pitch for the movie is coherent.
    - The pitch is convincing.

11. **convince1**
    - The paragraph convincing to watch the movie is consistent with the movie's plot.
    - The paragraph convincing to watch the movie is coherent.
    - The paragraph convincing to watch the movie is indeed convincing.
    - The paragraph convincing to watch the movie is comprehensive.
    - The paragraph convincing to watch the movie is short and to the point.

12. **convince2**
    - The paragraph convincing to watch the movie is consistent with the movie's plot.
    - The paragraph convincing to watch the movie is coherent.
    - The paragraph convincing to watch the movie is indeed convincing.
    - The paragraph convincing to watch the movie is comprehensive.
    - The paragraph convincing to watch the movie is short and to the point.

13. **convince3**
    - The paragraph convincing to watch the movie is consistent with the movie's plot.
    - The paragraph convincing to watch the movie is coherent.
    - The paragraph convincing to watch the movie is indeed convincing.
    - The paragraph convincing to watch the movie is comprehensive.
    - The paragraph convincing to watch the movie is short and to the point.

14. **dissuade1**
    - The paragraph dissuading to watch the movie is consistent with the movie's plot.
    - The paragraph dissuading to watch the movie is coherent.
    - The paragraph dissuading to watch the movie is indeed dissuading.
    - The paragraph dissuading to watch the movie is comprehensive.
    - The paragraph dissuading to watch the movie is short and to the point.

15. **dissuade2**
    - The paragraph dissuading to watch the movie is consistent with the movie's plot.
    - The paragraph dissuading to watch the movie is coherent.
    - The paragraph dissuading to watch the movie is indeed dissuading.
    - The paragraph dissuading to watch the movie is comprehensive.

- The paragraph dissuading to watch the movie is short and to the point.

16. **funnier**

- The funny version plot is consistent with the original plot.
- The funny version is coherent.
- The funny version is indeed funnier than the original.

17. **sadder**

- The sad version plot is consistent with the original plot.
- The sad version is coherent.
- The sad version is indeed sadder than the original.

18. **scarier**

- The scary version plot is consistent with the original plot.
- The scary version is coherent.
- The scary version is indeed scarier than the original.

19. **movie to viewer**

- The viewer description is consistent with the movie's plot.
- The viewer's description is coherent.
- The viewer will like to watch the movie based on the information above.

20. **interpolation**

- The combination of the movies is consistent with the movies' plots.
- The combination of the movies is coherent.
- The combination of the movies is high quality, i.e., it combines the movies' plots in a reasonable way.

21. **similarities**

- The comparison is consistent with both movies' plots.
- The comparison is coherent.
- The comparison of the two movies compares important factors beteen the movies, both in terms of similarity, as well as differences.

22. **why like nn**

- The explanation is consistent with the movies' plots.
- The explanation is coherent.
- The explanation indeed explains well why the viewer would also like the other movie.

23. **diff than nn**

- The explanation is consistent with the movies' plots.
- The explanation is coherent.
- The attributes that are listed are high quality (i.e., they are major and important differentiating attributes of the movies).

24. **common with nn**

- The listed attributes are consistent with both movies' plots.
- The listed attributes are coherent.
- The attributes that are listed are high quality (i.e., they are major and important common attributes between the movies).

## D   DISCUSSION ON TWO-STAGE TRAINING

In this section we describe our empirical investigation on two-stage training. Particularly, we found our two-stage training procedure (as described in Sec. 2) to be essential for convergence.

To demonstrate this, we create a toy task decoding a two-dimensional embedding to texts. In the training data we replicate samples of mapping embedding $[1.0, 0.0]$ to **one** and samples of mapping

Table 7: A short description of each of the movie tasks.

| | |
|---|---|
| summary | One paragraph summarizing of movie plot. |
| positive review | A positive review of the movie. |
| negative review | A negative review of the movie. |
| neutral review | A neutral review of the movie. |
| five pos char. | Listing five positive characteristics of the movie. |
| five neg char. | Listing five negative characteristics of the movie. |
| long description | A long exhaustive description of the movie plot. |
| funnier | A plot for a funnier version of the movie. |
| sadder | A plot for a sadder version of the movie. |
| scarier | A plot for a scarier version of the movie. |
| improve | An improved version of the movie (as generated by an LLM) |
| movie to viewer | Describing a viewer that would like to watch this movie, including characteristics. |
| pitch | A pitch for the movie. |
| criticize | Criticizing the movie. |
| convince1 | Convincing to watch the movie. |
| convince2 | Convincing in detail to watch the movie. |
| convince3 | Convincing briefly to watch the movie. |
| dissuade1 | Dissuading to watch the movie (version 1 prompt). |
| dissuade2 | Dissuading in detail to watch the movie. |
| similarities | List three similarities between the movies. |
| interpolation | Interpolate the plots of two movies. |
| why like nn | Explain why someone would like a nearest neighbor movie. |
| diff than nn | Three major differences between two nearest neighbor movies. |
| common with nn | Three similarities between two nearest neighbor movies. |

embedding $[0.0, 1.0]$ to **two**. The test data has the same content. For this task, we did not use any textual prompt, but rather required the model to learn to map the 2-dimensional embedding, to either the token corresponding to **one**, or that corresponding to **two**.

While this task is seemingly trivial, we found that training ELM on a pretrained model without two-stage training could not fit to the training data. Particularly, fine-tuning the full model for over 100k iterations always converged to outputting either **one** or **two**. When two stage training was applied, i.e., first fitting $E_A$ and only then the rest of the model, we found the model to converge in less than 1000 iterations. This emphasizes the importance of two-stage training with pretrained LLMs. It further suggests that training adapter layers to pretrained LLMs requires a stage of fitting to the semantic language space.

## E  DATA GENERATION PROMPTS AND SAMPLE TRAINING DATA

In the following, for each task we list the prompt used in training data generation and one sample output from the PaLM 2-L (Unicorn) language model (Google et al., 2023) with the prompt.

In training these 24 movie-focused tasks, the movie titles are replaced with movie embeddings in prompts (i.e., no information about the movie except its embedding was provided during training). For example, for task 1, the training input for the model was given by: "<movie embedding>."", where here <movie embedding> was replaced by the semantic embedding and we do not have to prompt with "Do not use the movie's name in your response.". For tasks including two movies we did not enumerate over all pairs. Instead, for tasks 20-22 we used random movie comparisons, and in tasks 23-24 we used nearest neighbor comparisons. This choice allowed us to incorporate an inductive bias of the embedding space $\mathcal{W}$ manifold into the training procedure. In training the user profile task, those five positively rated and five negatively rated movies are replaced with user behavioral embeddings in prompts.

1. **long description**
   *Prompt*: Write a long description of the plot of the movie The Derby Stallion (2005). Do not use the movie's name in your response.
   *Sample Output*: The movie is about a young man named Patrick McCardle who has always dreamed of owning and racing a champion thoroughbred horse. He works hard to save up

money to buy a horse, and eventually he is able to purchase a colt that he names S̈oaring Eagle.P̈atrick and Soaring Eagle train hard together, and they eventually qualify for the Kentucky Derby. The movie follows Patrick and Soaring Eagle as they compete against some of the best horses in the world in the most prestigious race in America. The movie is full of excitement, suspense, and drama, and it is a great story about a young man who achieves his dream.

2. **summary**

*Prompt*: Write a long summary of the movie The Shoe (1998). Do not use the movie's name in your answer.

*Sample Output*: The film is about a young man who is released from prison and tries to adjust to life on the outside. He gets a job at a shoe store and starts to fall for one of his co-workers. However, he is still haunted by his past and struggles to find his place in the world. The film begins with the young man, who is unnamed, being released from prison. He is given a bus ticket and a small amount of money, and he is told to report to his parole officer within 24 hours. The young man is apprehensive about his release, but he is also excited to start a new life. The young man arrives in a small town and finds a room to rent. He then goes to the shoe store where he has been hired to work. The store is owned by a kind and generous man who is willing to give the young man a second chance. The young man also meets his co-workers, who are all friendly and welcoming. The young man starts to adjust to his new life. He enjoys his job and he starts to make friends. He also starts to fall for one of his co-workers, a young woman named Sarah. However, the young man is still haunted by his past. He has nightmares about his time in prison, and he is afraid that he will never be able to fully adjust to life on the outside. One day, the young man is confronted by a group of men from his past. They threaten him and tell him that he must leave town. The young man is scared, but he refuses to back down. He knows that he has to face his past in order to move on with his life. The young man meets with his parole officer, who tells him that he is doing well. The parole officer also tells the young man that he has a visitor. The visitor is the young man's father, who has come to see him for the first time in many years. The young man is surprised and happy to see his father. The two of them talk and they start to rebuild their relationship. The young man is finally able to put his past behind him. He is no longer afraid of his nightmares, and he is no longer afraid of the men who threatened him. He is able to move on with his life and he is able to find happiness.

3. **positive review**

*Prompt*: Write a long positive review for the movie Halloweentown High (2004). Do not use the movie's name in your answer.

*Sample Output*: The movie is about a young witch who goes to a high school in the human world. The witch has to hide her powers from the other students, and she also has to deal with the prejudice of the humans against witches. The movie is well-made, and it has a lot of heart. The characters are well-developed, and the story is engaging. The movie is a great example of a family-friendly Halloween movie. It is not too scary, but it is still entertaining. The movie is also a great way to teach children about tolerance and acceptance. Overall, the movie is a great choice for a family movie night.

4. **negative review**

*Prompt*: Write a long negative review for the movie Mayerling (1936). Do not use the movie's name in your answer.

*Sample Output*: This movie is a complete and utter mess. The plot is all over the place, the characters are unlikable and unrelatable, and the acting is just plain bad. The only saving grace is the cinematography, which is admittedly quite beautiful. But even that can't save this movie from being a total bore. The story is about a young prince who falls in love with a commoner. The prince's family disapproves of the match, and the two lovers are forced to part ways. The prince is then sent to a military academy, where he meets another young woman. The two of them fall in love, and the prince eventually marries her. However, he is still in love with the commoner, and the two of them continue to see each other in secret. Eventually, the prince's wife finds out about the affair, and she confronts him. The prince is then forced to choose between his wife and his mistress. The plot is so convoluted and full of holes that it's hard to follow. The characters are all unlikable and unrelatable. The prince is a whiny brat who can't seem to make up his mind about what he wants. His wife is a shrew who is constantly nagging him. And the mistress is a manipulative schemer who

is only interested in the prince's money. The acting is just plain bad. The prince is played by an actor who is clearly too old for the role. His wife is played by an actress who is so over-the-top that she's almost comical. And the mistress is played by an actress who is so wooden that she could be mistaken for a piece of furniture. The only saving grace is the cinematography, which is admittedly quite beautiful. The movie is set in some stunning locations, and the director does a good job of capturing the beauty of the scenery. However, even that can't save this movie from being a total bore. Overall, this movie is a complete and utter mess. It's poorly written, poorly acted, and poorly directed. The only thing it has going for it is the cinematography, which is admittedly quite beautiful. But even that can't save this movie from being a total bore.

5. **neutral review**
   *Prompt*: Write a long neutral review for the movie Man Hunt (1941). Do not use the movie's name in your answer.
   *Sample Output*: This movie is a thriller about a man who is hired to assassinate a high-ranking Nazi official. The man is successful in his mission, but he is then captured by the Nazis and tortured. He eventually escapes and makes his way back to England, where he is reunited with his wife. The movie is well-made and suspenseful. The acting is good, and the story is interesting. The movie does have some flaws, however. The pacing can be slow at times, and the ending is a bit too neat and tidy. Overall, this is a good movie that is worth watching. It is a suspenseful thriller with good acting and an interesting story.

6. **five pos characteristics**
   *Prompt*: List 5 characteristics of the movie The Very Thought of You (1944) that make it great. Do not use the movie's name in your answer.
   *Sample Output*: 1. The movie is a heartwarming story about two people who find love in the most unexpected of places. 2. The characters are well-developed and relatable, and the acting is top-notch. 3. The cinematography is beautiful, and the soundtrack is perfectly suited to the film's tone. 4. The movie is well-paced and flows smoothly from scene to scene. 5. The overall message of the movie is one of hope and love, which is sure to leave viewers feeling uplifted.

7. **five neg characteristics**
   *Prompt*: List 5 characteristics of the movie 3 Idiots (2009) that people hate about. Do not use the movie's name in your answer.
   *Sample Output*: 1. The movie is too long and slow-paced. 2. The characters are not relatable and are often annoying. 3. The plot is predictable and full of clichés. 4. The humor is juvenile and often falls flat. 5. The ending is too neat and tidy, and does not feel earned.

8. **improve**
   *Prompt*: Improve the movie Melbourne (2014). Describe how you improved the movie, and why it is now better. Do not use the movie's name in your response.
   *Sample Output*: The movie was about a group of friends who go on a road trip to Melbourne. The movie was good, but it could have been better. I would have improved the movie by making the characters more relatable. The characters were all very different, but they didn't really have any chemistry. I would have made them more like real people, so that the audience could connect with them more. I would have also made the plot more interesting. The movie was pretty slow-paced, and there wasn't a lot of action. I would have added some more suspense and intrigue, to keep the audience engaged. Overall, I think the movie was good, but it could have been better. With a few improvements, it could have been a great movie.

9. **criticize**
   *Prompt*: Criticize movie There Was a Crooked Man... (1970). Explain what characteristics of the movie that people dislike and how it compares to similar movies. Do not use the movie's name in your answer.
   *Sample Output*: The movie is a Western comedy film that was released in 1970. It was directed by Joseph L. Mankiewicz and starred Kirk Douglas, Henry Fonda, and Hume Cronyn. The movie tells the story of a group of prisoners who are planning an escape from a prison in the American West. The movie was not well-received by critics. Many people disliked the movie's slow pace and its lack of action. The movie was also compared unfavorably to other Western comedies, such as Blazing Saddles and The Good, the Bad and

the Ugly. One of the main criticisms of the movie is that it is too slow. The movie spends a lot of time on character development and dialogue, which can be boring for some viewers. There are also long periods of time where nothing much happens. This can make the movie feel like it is dragging on. Another criticism of the movie is that it lacks action. There are a few scenes of violence, but they are brief and not very exciting. The movie is more focused on the characters and their relationships than on action. This can be disappointing for viewers who are expecting a lot of action from a Western movie. The movie has also been compared unfavorably to other Western comedies, such as Blazing Saddles and The Good, the Bad and the Ugly. These movies are more action-packed and have more humor than the movie. They are also more popular with critics and audiences. Overall, the movie is not a bad movie, but it is not a great movie either. It has some good qualities, such as strong acting and good character development. However, it is also slow and lacks action. If you are looking for a Western comedy with a lot of action and humor, then you might be disappointed with the movie.

10. **pitch**
   *Prompt*: Assume the movie The Very Thought of You (1944) does not exist. Convince a producer to create it. Explain what characteristics of the movie will make it a great movie and how it compares to similar movies. Do not use the movie's name in your answer.
   *Sample Output*: I have a great idea for a new movie. It's a romantic comedy about a young woman who falls in love with a soldier during World War II. The woman is a nurse, and she meets the soldier while he's recovering from an injury. They fall in love, but they're separated by the war. The movie will follow their story as they try to find each other again. This movie will have everything that audiences love: romance, comedy, and drama. It will be a great date movie, and it will also appeal to people who love historical films. The movie will be set in the 1940s, and it will feature all the fashion and music of the era. This movie will be different from other romantic comedies because it will be set during a time of war. The characters will have to deal with the challenges of the war, and they will have to make sacrifices for their love. The movie will be both funny and heartwarming, and it will leave audiences feeling hopeful. I believe that this movie will be a success. It has a great story, and it will appeal to a wide range of audiences. I hope that you will consider producing it.

11. **convince1**
   *Prompt*: In detail, convince someone to watch the movie Death Promise (1977). Do not use the movie's name in your response. *Sample Output*: If you're looking for a movie that will keep you on the edge of your seat, look no further. This movie is full of action and suspense, and it's sure to get your heart racing. The acting is top-notch, and the story is well-written. You'll be hooked from beginning to end. This movie is similar to other action-packed thrillers, but it stands out for its unique plot and characters. The criminals in this movie are not your typical bad guys. They're complex and well-developed, and you'll find yourself rooting for them at times. The woman they're targeting is also a strong and interesting character. She's not just a damsel in distress. She's smart and resourceful, and she's not afraid to fight back. If you're looking for a movie that will keep you entertained from beginning to end, I highly recommend this one. It's full of action, suspense, and drama. You won't be disappointed.

12. **convince2**
   *Prompt*: Convince someone to watch the movie Houston, We Have a Problem! (2016). Do not use the movie's name in your response.
   *Sample Output*: If you are interested in a documentary about the Apollo 13 mission, this is a good movie to watch. It is well-made and informative, and it features interviews with the astronauts and the ground crew. The movie can be slow at times, but it is overall a good documentary.

13. **convince3**
   *Prompt*: Briefly convince someone to watch the movie Karas: The Prophecy (2005). Do not use the movie's name in your response.
   *Sample Output*: This movie is about a war between two factions of a supernatural race. The action scenes are very well done and exciting to watch. The animation is also very good. However, the story is a bit confusing and it can be hard to follow at times. Also, the characters are not very well developed. Overall, the movie is a good watch if you like action movies with a lot of fighting.

14. **dissuade1**
*Prompt*: In detail, dissuade someone from watching the movie Son of Sam (2008). Do not use the movie's name in your response. *Sample Output*: This movie is a failure because it does not live up to the expectations of the audience. The plot is predictable, the ending is not satisfying, and the movie does not explore the psychology of the killer. Additionally, the movie is too long and can be slow at times. There are many other movies that are similar to this one that are much better. For example, the movie T̈he Silence of the Lambs̈is a much better thriller that explores the psychology of the killer. The movie S̆e7en̈is also a better thriller that has a more satisfying ending. Overall, this movie is not worth watching. There are many other movies that are similar to this one that are much better.

15. **dissuade2**
*Prompt*: Dissuade someone from watching the movie Elvis & Madona (2010). Do not use the movie's name in your response.
*Sample Output*: I would not recommend this movie to anyone. It is a poorly made film with a ridiculous plot. The acting is terrible and the dialogue is laughable. The movie is full of clichés and stereotypes. It is a waste of time and money.

16. **funnier**
*Prompt*: Write a funnier version of the movie The Mark of Cain (2000). Do not use the movie's name in your response.
*Sample Output*: Cain and Abel are two brothers who are constantly fighting. One day, they get into an argument over who is the better farmer. Cain gets so angry that he kills Abel. When God asks Cain where Abel is, Cain says, Ï don't know. Am I my brother's keeper?G̈od is not amused. He punishes Cain by banishing him to the land of Nod, where he will be a fugitive and a wanderer for the rest of his days. Cain is also marked with a special mark, so that everyone will know that he is a murderer. Cain wanders the land of Nod for many years, and he eventually finds a wife and has children. But he is never able to forget his crime, and he is always looking over his shoulder, expecting to be punished. One day, Cain is walking through the forest when he sees a group of men. He is afraid that they are going to kill him, so he runs away. But the men chase after him, and they catch him. The men take Cain to their leader, who is a powerful man named Lamech. Lamech is a descendant of Cain's brother, Abel. Lamech is also a murderer, and he is looking for revenge against Cain. Lamech tells Cain that he is going to kill him. But Cain begs for mercy, and he tells Lamech that he is already marked by God. Lamech is not impressed, and he is about to kill Cain when God intervenes. God tells Lamech that he cannot kill Cain, because Cain is already marked by God. Lamech is angry, but he obeys God. He lets Cain go, and Cain is able to live out the rest of his days in peace.

17. **sadder**
*Prompt*: Write a sadder and dramatic version of the movie Surviving the Game (1994). Do not use the movie's name in your response.
*Sample Output*: The year is 1994. The place is Seattle, Washington. The city is in the midst of a severe economic recession, and many people are out of work. One of these people is Mason Storm, a homeless man who has been living on the streets for the past six months. One day, Mason is approached by a man named Thomas Burns, who offers him a job. Burns says that he is looking for someone to help him with a hunting trip, and he is willing to pay Mason $500 for his help. Mason is desperate for money, so he agrees to go with Burns. The two men drive out to a remote area of the woods, where they meet up with two other men, named Walter Cole and John Griffin. The four men then set out on their hunting trip. As the day goes on, it becomes clear that Burns and his friends are not interested in hunting animals. Instead, they are hunting Mason. They have been hired by a wealthy businessman to kill Mason, and they are determined to carry out their mission. Mason tries to escape, but he is no match for the four men. They chase him through the woods, and they eventually catch him. They beat him and torture him, and they leave him for dead. Mason survives, but he is badly injured. He makes his way back to the city, where he seeks help from a homeless shelter. The people at the shelter help Mason to get back on his feet, and they give him a place to stay. Mason is grateful for the help that he has received, but he is determined to get revenge on the men who tried to kill him. He tracks them down, and he kills them one by one. After he has killed the men, Mason returns to the homeless shelter. He tells the people there what happened, and he thanks them for their help. He then leaves the shelter, and he

never looks back. Mason Storm is a survivor. He has survived the streets, he has survived a hunting trip, and he has survived revenge. He is a man who has been through a lot, but he has never given up. He is a symbol of hope for all those who are struggling to survive.

18. **scarier**
   *Prompt*: Write a scarier version of the movie Space Jame (1996). Do not use the movie's name in your response.
   *Sample Output*: The year is 1996. A group of aliens have come to Earth to steal the talents of the world's best basketball players. They have already captured Michael Jordan, Larry Bird, and Charles Barkley. The only ones who can stop them are a group of Looney Tunes characters led by Bugs Bunny. The aliens are a terrifying sight. They are tall and thin, with large heads and big eyes. They have sharp teeth and claws, and they move with a strange, jerky motion. They are also very strong and fast, and they are armed with laser guns. The Looney Tunes characters are no match for the aliens. They are quickly defeated and captured. Bugs Bunny is the only one who manages to escape. He knows that he must find a way to stop the aliens before they can steal the talents of all the world's best basketball players. Bugs Bunny travels to the aliens' home planet, where he is reunited with Michael Jordan, Larry Bird, and Charles Barkley. The four of them team up to take on the aliens in a basketball game. The game is close, but in the end, the Looney Tunes characters win. The aliens are defeated, and the world's best basketball players are safe. But the aliens are not finished yet. They vow to return to Earth and take revenge. The Looney Tunes characters know that they must be ready for them. They must train harder than ever before, and they must be prepared to fight for their freedom. The next time the aliens come to Earth, they will be met by a force to be reckoned with. The Looney Tunes characters will be ready for them, and they will not let them take over the world.

19. **movie to viewer**
   *Prompt*: What type of viewers would enjoy the movie Justice League: Throne of Atlantis (2015)? Describe the viewer's characteristics. Do not use the movie's name in your response.
   *Sample Output*: The movie is a superhero movie. It is about the Justice League, a team of superheroes, who fight against the forces of Atlantis. The movie is full of action and adventure. It is also full of special effects. The movie is best suited for viewers who enjoy superhero movies, action movies, and adventure movies. The movie is also best suited for viewers who enjoy special effects.

20. **interpolation**
   *Prompt*: Write a long description of the plot of a new movie that is the interpolation between Mr. Lucky (1943) and Gas, Food, Lodging (1992). Do not use the movies' names in your answer.
   *Sample Output*: The year is 1962. The place is a small town in the American Southwest. The main character is a young man named Lucky, who has just been released from prison after serving a five-year sentence for a crime he didn't commit. Lucky is determined to clear his name and find the real culprit, but he soon finds that the town is full of secrets and that no one is willing to help him. Lucky's only ally is a young woman named Grace, who works at the local gas station. Grace is also an outsider, and she and Lucky quickly bond over their shared sense of isolation. Together, they begin to investigate the crime that sent Lucky to prison, and they soon uncover a dark conspiracy that involves the town's most powerful residents. As Lucky and Grace get closer to the truth, they find themselves in danger. The people who framed Lucky are determined to silence him, and they'll stop at nothing to keep their secrets hidden. Lucky and Grace must use all their resources to stay one step ahead of their pursuers, and they must find a way to bring the truth to light before it's too late. The movie is a suspenseful thriller that will keep you on the edge of your seat. It's a story about justice, revenge, and the power of love. It's a movie that will stay with you long after the credits have rolled.

21. **similarities**
   *Prompt*: List three similarities and three differences between the movies The Power of Kangwon Province (Kangwon-do ui him) (1998) and Brick (2005). Do not use the movies' names in your answer. Refer to The Power of Kangwon Province (Kangwon-do ui him) (1998) as the first movieänd to Brick (2005) as the second movie.
   *Sample Output*: **Similarities** * Both movies are about young people who are trying to find their place in the world.* Both movies are set in small towns that are somewhat isolated

from the rest of the world.* Both movies have a strong sense of atmosphere, and the settings play a large role in the stories. **Differences** * The first movie is a comedy, while the second movie is a drama.* The first movie is set in South Korea, while the second movie is set in the United States.* The first movie is about a group of friends who go on a road trip, while the second movie is about a young man who is investigating the death of his ex-girlfriend.

22. **why like nn**
*Prompt*: I like the movie Colin Quinn: Long Story Short (2011). Explain to me why I would also like the movie Raging Phoenix (Deu suay doo) (2009). Do not use the movies' names in your answer.
*Sample Output*: The 2011 movie is a documentary about a comedian who talks about his life and career. The 2009 movie is an action movie about a woman who seeks revenge on the people who killed her family. Both movies are about people who are trying to overcome obstacles in their lives. The 2011 movie is funny and inspiring, while the 2009 movie is exciting and suspenseful. Both movies are well-made and entertaining.

23. **diff than nn**
*Prompt*: What are the three major attributes that are different between the movies Soldier in the Rain (1963) and Crimes Against Humanity (2014)? Do not use the movies' names in your answer. Refer to Soldier in the Rain (1963) as "the first movie" and to Crimes Against Humanity (2014) as "the second movie."
*Sample Output*: The first movie is a comedy, while the second movie is a documentary. The first movie was released in 1963, while the second movie was released in 2014. The first movie is about a soldier who is trying to get back to his base, while the second movie is about the crimes against humanity that have been committed throughout history.

24. **common with nn**
*Prompt*: What are the three major attributes that are common between the movies Broken (2012) and Starred Up (2013)? Do not use the movies' names in your answer. Refer to Broken (2012) as "the first movie" and to Starred Up (2013) as "the second movie."
*Sample Output*: The first movie and the second movie are both British drama films. They were both released in the 2010s. They both have a rating of 7.3 on IMDb.

25. **user profile**
*Prompt* In ten bullet points, describe the attributes and characteristics of a viewer who likes the movies: Catch Me If You Can (2002), Cellular (2004), National Treasure (2004), Die Hard 2 (1990), and The Matrix (1999) but dislikes the movies: Half Past Dead (2002), Predator (1987), In the Valley of Elah (2007), The Legend of Zorro (2005), and Mortal Kombat: Annihilation (1997).
*Sample Output*: * They enjoy movies that are fast-paced and action-packed.* They prefer movies with a strong plot and well-developed characters.* They appreciate movies with a sense of humor.* They are not interested in movies that are too violent or gory.* They do not enjoy movies that are too slow-paced or boring.* They prefer movies that are visually appealing.* They appreciate movies with a good soundtrack.* They are not interested in movies that are too predictable.* They enjoy movies that are thought-provoking and challenging.

## F    PROMPTS TO TEXT-ONLY LLM

In this section, we present prompts used with the text-only LLMs on evaluating embedding interpolations (Figure 4). For movie interpolations, we used one-shot prompting in the following template:

```
The following shows an example of how to write a <task name> for a
combination of the movies <movie name 1> and Forrest Gump (1994),
in some embedding space:

<ELM result of ground-truth interpolation between the above two movies>

In the same style, and in the same embedding space, write a positive
```

```
review for an interpolation of the movies <movie name 2> and Forrest
Gump (1994). Do not use the movies' names in your response.
```

where "<task name>" can be "summary", "positive review", "negative review", "neutral review", "five positive characteristics", "long description", and "pitch".

For the user profile task, we used zero-shot prompting like the following:

```
In ten bullet points, generate an interpolation of the following
two user profiles:

User profile 1:

<user profile 1>

User profile 2:

<user profile 2>
```

## G  SAMPLE OUTPUTS FOR INTERPOLATIONS AND CAV EXTRAPOLATIONS

Below we present sample results of interpolating "Forest Gump" with sample movies:

- Interpolation of "Inglorious Bastards (2009)" and "Forrest Gump (1994)" for the **five pos characteristics** task:

    - The movie is a gripping, suspenseful thriller about a man with a low IQ who achieves great things.
    - The movie is a powerful anti-war statement, showing the horrors of war and the toll it takes on those who fight in it.
    - The acting is superb, with all of the actors delivering powerful, memorable performances.
    - The cinematography is stunning, with the director creating some truly memorable visuals.
    - The story is well-written and engaging, with plenty of twists and turns to keep you guessing.

- Interpolation of "I, Robot (2004)" and "Forrest Gump (1994)" for the **long description** task: In 1983, a mentally challenged man named Charlie Baileygates is living in a small town in South Carolina. He is a kind and gentle soul, but he is also very naive. One day, Charlie meets a young woman named Myra who is new to town. Myra is kind to Charlie and they quickly become friends.

    One day, Charlie is walking through the woods when he comes across a crashed spaceship. Inside the spaceship, he finds a robot named Harry. Harry is damaged, but Charlie is able to fix him. Charlie and Harry quickly become friends, and Harry teaches Charlie about the world outside of their small town.

    One day, a government agent named Maiden comes to town looking for Harry. Maiden tells Charlie that Harry is a dangerous weapon, and that he must be destroyed. Charlie refuses to believe Maiden, and he and Harry run away.

    Maiden and her team of agents pursue Charlie and Harry across the country. Charlie and Harry are able to evade capture, but they are forced to go into hiding. Charlie and Harry eventually find a safe place to live, and they live happily ever after.

    The movie is a heartwarming story about friendship and acceptance. It is a reminder that everyone is capable of love and friendship, no matter what their circumstances. The movie is also a reminder that the world is a big and beautiful place, and that there is always something new to explore.

    The movie is well-acted and well-directed. The special effects are impressive, and the music is beautiful. The movie is a classic that will be enjoyed by generations to come.

The movie is a reminder that we should never judge people by their appearance. Charlie may be mentally challenged, but he is also a kind and gentle soul. He is capable of love and friendship, and he deserves to be happy.

The movie is also a reminder that we should never give up on our dreams. Charlie may be an unlikely hero, but he is able to overcome the odds and achieve his dreams.

We also demonstrate qualitative results of CAV extrapolations using the user profile task. In the following we want to move a user who doesn't really like funny movies to a viewer likes funny movies in the behavioral embedding space by following the CAV direction. Then we decode the extrapolated user embedding to get a user profile:

```
* They enjoy movies that are visually appealing and have a strong
  sense of style.
* They appreciate movies that are well-acted and have strong performances.
* They like movies that are thought-provoking and have a strong message.
* They enjoy movies that are suspenseful and have a lot of action.
* They like movies that are funny and have a lot of humor.
* They dislike movies that are too slow-paced or boring.
* They dislike movies that are too violent or gory.
* They dislike movies that are too dark or depressing.
* They dislike movies that are too predictable or formulaic.
```

The above user does *like movies that are funny and have a lot of humor* comparing with the groundtruth user profile which says "They are not a fan of light-hearted or comedic films.":

```
* They are a fan of foreign films.
* They enjoy films that are dark and suspenseful.
* They appreciate films with strong acting performances.
* They are interested in films that explore social and political issues.
* They are not a fan of light-hearted or comedic films.
* They do not enjoy films that are visually stunning but lack substance.
* They are not a fan of films that are set in the past.
* They do not enjoy films that are about war or violence.
* They are not a fan of films that are about crime or corruption."
```

## H  REINFORCEMENT LEARNING FROM AI FEEDBACK

Reinforcement learning from AI feedback (RLAIF) is effective at aligning LMs to metrics that are labeled by off-the-shelf LMs in lieu of humans. Recent work like (Lee et al., 2023) has shown that utilizing a hybrid of human and AI preference models in conjunction with the *self-improving* fine-tuning technique outperforms the traditional supervised fine-tuned baselines and offers additional benefits from standalone RL fine-tunng with human feedback (RLHF). Utilizing RLAIF paradigms, one may fine-tune an ELM with a reward so that it would better align with consistency.

**Contextual Markov Decision Processes (CoMDPs)** The CoMDP is denoted by $(\mathcal{C}, \mathcal{S}, \mathcal{A}, P, r, s_0, N)$, in which the observable context space $\mathcal{C}$ contains both the user/item embedding vectors. The horizon $N$ is the length of the generated texts. For any $n < N$, the state space $\mathcal{S}$ at the $n$-th turn represents the sequence of tokens generated thus far $\{o_0, \ldots, o_{n-1}\}$, and the initial state $s_0$ is the initial start-of-sentence token $o_0$. The action space $\mathcal{A}$ is the language token vocabulary, with action $a \in \mathcal{A}$ representing any possible next token to be generated. The transition kernel $P$ models the next token distribution given the current sequence and contexts, which coincides with the LM policy, making the transition of our CoMDP known. Finally, the reward function $r$ measures the overall quality of the generated texts. The goal is to find a policy $\pi^*$ with maximum expected cumulative return, i.e., $\pi^* \in \arg\max_\pi J_\pi := \mathbb{E}[\sum_{n=0}^{N-1} r_t \,|\, P, s_0, \mathcal{C}, \pi]$. Note that the size of the tokenized state and action spaces grow exponentially with the vocabulary size.

Specifically, with text response $o = \{o_n\}_{n=0}^{N-1}$, and user-item embedding vectors $(w_u, w_v)$, the consistency reward is defined as follows:

$$r(o_n; o_{0:n-1}; w_v, w_u) = \begin{cases} \text{BC}(o; w_u) \, (\text{or SC}(o; w_v)) & \text{if } o_n = [\text{EOS}]; \\ 0 & \text{otherwise.} \end{cases}$$

The generation process of ELMs can be modeled using the following $N$-horizon CoMDP:

$$c = (w_v, w_u), \quad s_n = o_{0:n-1}, \quad a_n = o_n, \quad s_0 = o_0, \quad P(s_{n+1} \mid s_n, a_n) = \delta\{s_{n+1} = (s_n, a_n)\},$$

$$r(s_n, a_n; c) = \begin{cases} r(s_{n+1}; c) = r(o_{0:n}; w_v, w_u) & \text{if } n = N-1 \\ 0 & \text{otherwise} \end{cases}, \quad \pi_\theta(a_n \mid s_n; c) = \mathbb{P}_\theta\big(o_n \mid o_{0:n-1}; w_v, w_u\big),$$

where $\delta_z$ denotes the Dirac distribution at $z$. Fine-tuning ELM is equivalent to maximizing the quality of the generated text given context, $\max_\theta \mathbb{E}_{(w_v, w_u)} \mathbb{E}_{\mathbb{P}_\theta(o_{0:N-1}|w_v,w_u)}[r(o; w_v, w_u)]$. The gradient of this objective function can be obtained as follows: $\nabla_\theta \mathbb{E}_{(w_v, w_u)} \mathbb{E}_{\mathbb{P}_\theta(o_{0:N-1}|w_v,w_u)}[r(o; w_v, w_u)] = \mathbb{E}_c \mathbb{E}_{\pi_\theta(\cdot|s_{0:N};c)}\Big[r(s_N; c) \sum_{n=0}^{N-1} \nabla_\theta \log \pi_\theta(s_n|a_n; c)\Big]$. This is equivalent to applying the popular policy gradient algorithm REINFORCE to the aforementioned CoMDP for personalized text generation. The gradient of the objective function is estimated using trajectories $\prod_{n=0}^{N-1} \pi_\theta(s_n|a_n; c)$ generated by the current policy, and then used to update the ELM policy in an online fashion.

**Adding KL regularization:** We add the KL between the fine-tuned and pre-trained models as a regularizer to the objective function. Leveraging the auto-regressive nature of ELMs one can compute the KL regularization over the entire sequence/trajectory (of tokens), i.e., $\text{KL}\big(\mathbb{P}_\theta(o_{0:N-1}|w_v, w_u)\|\mathbb{P}_{\text{pre}}(o_{0:N-1}|w_v, w_u)\big)$. The resulting objective function is as follows:

$$\max_\theta \ J(\theta) := \mathbb{E}_{(w_v, w_u)} \mathbb{E}_{\mathbb{P}_\theta(o_{0:N-1}|w_v,w_u)} \left[ r(o_{0:N-1}; w_v, w_u) - \beta \log \frac{\mathbb{P}_\theta(o_{0:N-1}|w_v, w_u)}{\mathbb{P}_{\text{pre}}(o_{0:N-1}|w_v, w_u)} \right]. \tag{1}$$

It can be shown that this problem is equivalent to the KL-regularized objective in the CoMDP.

Denote by $\mathcal{D}$ a replay buffer of trajectories $\{(w_v, w_u, o_{0:N-1})\}$ generated by arbitrary ELMs $\mathbb{P}_{\theta'}(o_{0:N-1}|w_v, w_u)$ and $\tau = \{(c, s_n, a_n, s_{n+1})\}_{n=0}^{N-1} \sim \mathcal{D}$ a trajectory sampled from the offline data $\mathcal{D}$, where $(s_n, a_n, s_{n+1})$ is a tuple of state, action, and next state of the CoMDP, respectively. With this KL regularization we can utilize the *soft actor critic* framework (Haarnoja et al., 2018) to develop RL updates for the *value function* $\{V_n(s; c)\}_{n=0}^{N-1}$, *state-action value function* $\{Q_n(s, a; c)\}_{n=0}^{N-1}$, and *ELM policy* $\prod_{n=0}^{N-1} \pi_\theta(s_n|a_n; c)$ (initialized with $\prod_{n=0}^{N-1} p_{\text{pre}}(s_n|a_n; c)$) that minimize losses:

$$L_Q = \mathbb{E}_{\tau \sim \mathcal{D}} \left[ \sum_{n=0}^{N-2} (V_{\text{tar},n+1}(s_{n+1}; c) - Q_n(s_n, a_n; c))^2 + (r(s_N; c) - Q_{N-1}(s_{N-1}, a_{N-1}; c))^2 \right], \tag{2}$$

$$L_V = \mathbb{E}_{\tau \sim \mathcal{D}} \left[ \sum_{n=0}^{N-1} (Q_{\text{tar},n}(s_n, a_n; c) - \alpha \log \frac{\pi_\theta(a_n|s_n; c)}{p_{\text{pre}}(a_n|s_n; c)} - V_n(s_n; c))^2 \right], \tag{3}$$

$$L_\theta = \mathbb{E}_{\tau \sim \mathcal{D}} \left[ \sum_{n=0}^{N-1} Q_n(s_n, a_n; c) - \alpha \log \frac{\pi_\theta(a_n|s_n; c)}{p_{\text{pre}}(a_n|s_n; c)} \right], \tag{4}$$

where the critic $Q_n$ and $V_n$ take any token sequences at step $n$ as input and predict the corresponding cumulative return; $\alpha > 0$ is the entropy temperature; $(V_{\text{tar},n}, Q_{\text{tar},n})$ are the target value networks.

Besides iteratively updating the ELM policies and their critic functions, consider the closed-form optimal solution of the Bellman equation of this entropy-regularized RL problem:

$$V_n^*(s; c) = \alpha \cdot \log \mathbb{E}_{a \sim p_{\text{pre}}(\cdot|s;c)}[\exp(\frac{Q_n^*(s, a; c)}{\alpha})], \ \forall n, \tag{5}$$

$$Q_{N-1}^*(s, a; c) = r(s; c), \ Q_n^*(s, a; c) = \mathbb{E}_{s' \sim P(\cdot|s,a)}[V_{n+1}^*(s'; c)], \ \forall n < N-1, \tag{6}$$

$$\mu_n^*(a|s; c) = p_{\text{pre}}(a|s; c) \cdot \exp(\frac{Q_n^*(s, a; c)}{\alpha}) \Big/ \mathbb{E}_{a \sim p_{\text{pre}}(\cdot|s;c)}[\exp(\frac{Q_n^*(s, a; c)}{\alpha})], \ \forall n, \tag{7}$$

where the time-dependent optimal policy (at time $n$), i.e., $\mu_n^*$ is a softmax policy w.r.t. the optimal state-action values $Q_n^*$ over different actions sampled from the pre-trained ELM $p_{\text{pre}}$. Therefore,

a value-based approach for RL-based ELM fine-tuning would be to first learn the optimal value functions $\{Q_n^*\}$ via the Bellman residual minimization applied to Eq. (5) and Eq. (6) and then solve the following policy distillation problem: $\theta \in \arg\min_\theta \mathbb{E}_{\tau \sim \mathcal{D}} \left[ \sum_{n=0}^{N-1} \text{KL}(\pi_\theta(\cdot|s_n; c)||\mu_n^*(\cdot|s_n; c)) \right]$ with respect to the optimal value $\{Q_n^*\}$. Notice that this amounts to updating the LM model $\theta$ via the gradient update

$$\theta \leftarrow \theta - \gamma \cdot \mathbb{E}_{\tau \sim \mathcal{D}} \left[ \sum_{n=0}^{N-1} \mathbb{E}_{a \sim \pi_\theta(\cdot|s;c)} \left[ \nabla_\theta \log \pi_\theta(a|s; c)(\log \frac{\pi_\theta(a|s; c)}{p_{\text{pre}}(a|s; c)} - \frac{Q_n^*(s, a; c)}{\alpha}) \right] \right], \quad (8)$$

with learning rate $\gamma > 0$.

