# OpenReview forum: "Demystifying Embedding Spaces using Large Language Models"
_ICLR.cc/2024/Conference — ICLR 2024 poster_

### Official Review · Reviewer_viWv · 2023-10-31

**Soundness:** 3 good
**Presentation:** 3 good
**Contribution:** 3 good
**Rating:** 8
**Confidence:** 4

**Summary:**

The authors propose a framework (ELM) for interpreting the vectors in domain-embedding spaces using text-only LLMs. The main idea is to project vector representations and textual tokens to a conjoint vector space. This is achieved by training adapter layers that serve as an interface to the language model. This adapter layers, together with the pre-trained model, create a new, extended model that maps both tokens and embeddings (from a arbitrary latent metric space) into a common space. The authors put this framework to the test by attempting to interpret two different semantic embeddings that they generate from the MovieLens dataset: One that models the users by their ratings of movies and one  that models the movies themselves based on their descriptions. The input data for the ELM framework then consists, for example, of text sequences in which the title of a movie has been replaced by the correspondingly generated semantic embedding. In order to verify the results of the framework, the authors rely on the expertise of test persons on the one  hand, and on the other hand, they also design two metrics to test for consistency.

**Strengths:**

I think the strengths of the article lie in the idea of extending a model to accept sequential tokens and embeddings as input data. I agree with the authors that this approach has the potential to provide a tool to study corresponding vector spaces spanned by semantic representations. In addition, this offers interesting possibilities for applications such like recommendation systems. Based on the elaborate and convincing experiments, the authors can very well prove the merits of their approach. Another positive feature is the detailed appendix.

**Weaknesses:**

Unfortunately, in my opinion, the actual core of the article is somewhat overshadowed by the detailed experiments. I would also have liked to see a figure (in addition to Figure 2) that shows the structure of the framework at a glance. In my opinion, the surface planes shown in the 3D graphs of Figure 1 do not contribute much to the clarity either, since they only exemplify a distance of points in the vector space. The authors take effords to show the benefits of their framework with two examples (movie ratings - behavioral embeddings and movie descriptions - semantic embeddings), but perhaps one of the vector spaces studied could have come from a completely different domain, perhaps derived from a different dataset. But this probably falls into future work. On the whole, the paper is very well structured and easy to read, but the section on related work seems a bit out of place before the section conclusions. Perhaps it could have been placed further forward in the text.

**Questions:**

•	Has the framework been tested on other embeddings that may contain less semantic information than the domain embeddings used here (similar to the example in Appendix C)?
•	How do the authors assess the potential usefulness of their approach for the interpretation of other dense vector representations, for example graph or image embeddings?

---

> ### Author Response · Authors · 2023-11-14
> **Author Response**
>
> Thank you for your positive review and feedback. We appreciate the fact that you recognized the potential of ELM as a tool for studying embedding spaces. Below are some brief responses to the points you raise.
>
> We added a figure showing the structure of the framework (see the updated version of Figure 2), which we hope will clarify our approach—thanks for the suggestion.
>
> Also, thank you for your comments regarding the embeddings. Notice that our behavioral embeddings are trained solely on user/movie ID and rating data (with no semantic information). Our user profiles task successfully demonstrates the ability of extracting semantic information from behavioral embeddings. Hope that addresses the concern regarding generalizability on embedding information from one domain to another. To emphasize this further, we are also adding experiments on Amazon public data, where the embeddings similarly come from a different source. We hope this will answer your concern.
>
> Thank you again for your positive review and feedback. Please let us know if there are any more concerns we can address.

---

> > ### Author Response · Authors · 2023-11-18
> > **Additional Experiments on the Amazon Dataset**
> >
> > We've added new experiments for ELM using six tasks over Amazon dataset (see Appendix A in the updated version of the paper). Please let us know if there are further concerns we can address.

---

### Official Review · Reviewer_bEK6 · 2023-10-31

**Soundness:** 3 good
**Presentation:** 2 fair
**Contribution:** 2 fair
**Rating:** 6
**Confidence:** 2

**Summary:**

This paper presents the Embedding Language Model (ELM), a novel language model framework for interpreting domain-embedding spaces. By inserting trainable adapters into existing large language models (LLMs), ELM can accept domain embedding vectors as parts of its textual input sequence to allow the interpretation of continuous domain embeddings using natural language. Abundant experiments demonstrate ELM's proficiency in understanding, navigating, and manipulating complex embedding representations.

**Strengths:**

- Interpreting abstract embeddings into human-understandable natural language descriptions is intuitively appealing.
- The proposed approach is simple and effective by reasonably leveraging the power of large language models (LLMs).
- The authors have comprehensively assessed the quality of ELM's outputs using a variety of evaluation techniques, including qualitative human evaluations and specific consistency metrics.

**Weaknesses:**

Honestly, one of my greatest concerns is the practicality of the proposed framework, given that training an ELM requires manually constructing a batch of tasks, which need to be diverse enough to extract rich semantic information from the target domain $\mathcal{W}$ to support the interpretation of the embeddings. Admittedly, the experimental part of the paper validates ELM's proficiency in interpreting two forms of embeddings on a movie-related dataset. However, I am not sure if ELM performs equally well in other more specialized domains, considering that the original training corpus of LLMs is likely to contain rich semantic information relevant to movies. In addition, constructing diverse task prompts may be tedious for a realistic user, so are the prompts presented in Appendix D basically applicable to other domains? After all, the most straightforward need for a user is to know what an embedding represents in general.

- Given the prevalence of adapter tuning [1], there is nothing new to me in the method.

1. Parameter-efficient transfer learning for NLP, Neil Houlsby et al.

**Questions:**

- What is the detailed form of the loss function $\mathcal{L}\_{\theta}$ used in the experiment? Did the authors use reinforcement learning from AI feedback to optimize $\mathcal{M}_{ELM}$?
- Since the output of instances in the training data is generated by LLMs, can ELM reasonably interpret embeddings in specialized domains outside the scope of LLMs?

**Details Of Ethics Concerns:**

No ethical issues found.

---

> ### Author Response · Authors · 2023-11-14
> **Author Response**
>
> Thank you for your helpful comments. We are happy you found the problem of interpreting embeddings using language appealing, and our approach simple and effective.
>
> We understand your concern w.r.t. ELM, which requires the availability of tasks mapping the domain embeddings to text. While some problems would indeed require human intervention for the creation of such tasks (or possibly using the power of existing LLMs, as we’ve shown in the paper), many tasks readily associate enormous amounts of text with specific entities. Users and items in recommender systems, as an example, are abstract entities which are often described using behavioral embeddings, which are hard to interpret. Nevertheless, users also engage with text and items by commenting, reviewing, and conversing with other users about items (e.g., Youtube comments, Amazon reviews, MovieLens tags, etc.).
>
> To address your concern, we are adding experiments for the large public Amazon dataset [1], which consists of 9.35M items with textual descriptions, 20.9M users, 233.1M reviews, and 82.83M ratings. We are training ELM on 5-7 tasks over this dataset (starting with item descriptions, positive reviews, negative reviews, neutral reviews, convincing item summaries, and user profiles), and will update our paper with these new results by the end of the week, including human evaluation. We hope this will address most of your concerns.
>
> Regarding your questions:
> 1. We used supervised learning (i.e., next token cross-entropy loss) to train all our models in the paper. In Appendix G, we discuss training ELM using reinforcement learning from AI feedback (RLAIF). Specifically, we attempt to train ELM with an additional reward to encourage semantic and behavioral consistency. Our results, however, did not show significant improvement compared to the regular supervised learning baseline. We leave further experimentation using different RLAIF approaches for future work.
> 2. To address your second question, we are adding new results for the Amazon product dataset (as discussed above).
>
> Reference for the Amazon Dataset: \
> [1] McAuley, Julian, Christopher Targett, Qinfeng Shi, and Anton Van Den Hengel. "Image-based recommendations on styles and substitutes." In Proceedings of the 38th international ACM SIGIR conference on research and development in information retrieval, pp. 43-52. 2015.

---

> > ### Author Response · Authors · 2023-11-18
> > **Additional Experiments on the Amazon Dataset**
> >
> > We've added new experiments for ELM using six tasks over Amazon dataset (see Appendix A in the updated version of the paper). We hope these experiments address your main concerns. Please let us know if there are further concerns we can address.

---

> > > ### Comment · Reviewer_bEK6 · 2023-11-22
> > > **Thanks for the response**
> > >
> > > I appreciate the authors' comprehensive response, and I think the added experiments on the large public Amazon dataset and further human evaluation will improve the quality of this paper. My concerns have been largely addressed, so I raised my score from 5 to 6.

---

### Official Review · Reviewer_pa1w · 2023-10-31

**Soundness:** 3 good
**Presentation:** 3 good
**Contribution:** 3 good
**Rating:** 8
**Confidence:** 3

**Summary:**

This paper presents ELM, a framework that uses LLM to interpret embeddings. By training an adapter, ELM maps the domain embedding space to the LLM token embedding space, allowing users to use natural language to query and understand the domain embedding space. An evaluation of 24 original tasks with a movie rating dataset shows that ELM provides high semantic and behavioral consistency. Finally, the paper shows promising results of using ELM to query hypothetical embeddings and generalize concept activation vectors.

**Strengths:**

- [S1] The proposed method ELM is novel and intuitive. The two-stage training (adapter first, full model next) is interesting and makes sense.
- [S2] I appreciate including human evaluation in section 3. The analysis and discussion on hypothetical embedding vectors and concept activation vectors (section 4) are very interesting.
- [S3] The paper overall is well-written and easy to follow.

**Weaknesses:**

## Major weaknesses

- [W1] The evaluation can be improved. (1) The training data are all synthesized from an LLM. (2) The 24 tasks are all original. (3) ELM is only evaluated on the MovieLens. (4) There is no baseline method. It is unclear how well ELM will perform on real data and tasks compared to other methods.

## Minor weaknesses

- [M1] It would be great to explain $E_A$ in Figure 2 (the definition is on page 4).
- [M2] The 24 tasks are essential to understand this paper. I recommend adding a table to provide a high-level description of these 24 tasks in the main paper.
- [M3] The ELM method is quite intuitive, but the writing in section 2 is overly-complex. It would be helpful to have a small glossary to explain all notations.
- [M4] It is hard to make sense of Table 1. Are these numbers good?

**Questions:**

- [Q1] Have you tried ELM on other datasets and tasks other than movie reviews?
- [Q2] Figure 1 is a bit hard to understand. For example, how do people generate the embedding for the animated version of Forrest Gump? Interpolating the embedding of Forrest Gump with an animated movie? Is the black text output from an LLM? Who writes the blue text (e.g., users, researchers)?

---

> ### Author Response · Authors · 2023-11-14
> **Author Response**
>
> Thank you for your positive feedback and fruitful comments. We appreciate that you found our method novel and intuitive, and the discussion in the paper interesting. Please find our response below.
>
> To address your main concern, we are adding new tasks on real data from the public Amazon product dataset [1], which consists of 9.35M items with textual descriptions, 20.9M users, 233.1M reviews, and 82.83M ratings. We are training tasks involving item descriptions, positive reviews, negative reviews, neutral reviews, convincing item summaries, and user profiles. We will update the paper with these results by the end of the week, and also include human rater evaluations to improve the evaluation of our paper. We hope this will address your main concern.
>
> Thank you also for your other suggestions, based on which we’ve updated the paper as follows (changes marked in red in the paper): \
> [M1] We’ve added an explanation for $E_A$ in Figure 2. \
> [M2] We’ve added a short description of each task to the paper (see Table 3) \
> [M3] We’ve added a small glossary to explain our notation, which we hope will improve clarity. (see Table 1) \
> [M4] For human rater evaluations, the numbers correspond to ratings between “agree” and “strongly agree”. For semantic and behavioral consistency metrics, we show in Figure 3 attempts of using GPT4 and PaLM2 text-only baselines, with significantly lower scores.
>
>
> Regarding your questions: \
> [Q1] As mentioned above, we will add experiments using the public Amazon dataset. \
> [Q2] To generate a new movie (item) embedding, we found two potential options: \
> Option 1: One could create an item embedding of a hypothetical movie for which the specific attribute is most salient (whereas the other attributes are average). This movie does not necessarily need to exist. \
> Option 2: Moving in the direction of the “concept activation vector” with respect to a specific movie soft attribute (e.g., violent, funny), as we demonstrate with user profiles.
>
> In the paper we demonstrate Option 1 for movie tasks and Option 2 for the user profile task.
>
> Finally, regarding the text in Figure 1:
> The black text is the generated output of ELM.
> The blue and red text are the prompts used as input to ELM, where the red part is an embedding vector not text.
> For instance if Forrest Gump’s movie embedding is [1.0, 1.0] and the direction of funny is [0, 1], then the prompt could be of the following sort: “List five positive characteristics of the movie [1.0, 1.1]”, where here, the vector [1.0, 1.1] is inputted through the adapters of ELM, and not as text.
> We hope this clarifies your question.
>
> Reference for the Amazon Dataset: \
> [1] McAuley, Julian, Christopher Targett, Qinfeng Shi, and Anton Van Den Hengel. "Image-based recommendations on styles and substitutes." In Proceedings of the 38th international ACM SIGIR conference on research and development in information retrieval, pp. 43-52. 2015.

---

> > ### Comment · Reviewer_pa1w · 2023-11-14
> > **Thank you for the response!**
> >
> > Thank you for the quick and thorough response! All my concerns have been addressed. Some final minor "style" suggestions:
> >
> > 1. In the glossary table, "pretrained LLM" $\rightarrow$ "Pretrained LLM"
> > 2. It would be great if you can incorporate your response to [M4] to the table caption and highlight important number in the table. The table is currently a "block of numbers" —it is hard to know what is the take-away message.
> > 3. Regarding Q2, it would be great if you can explain the color encodings (e.g., through annotations in the figure or explaining it in the caption).
> >
> > Good work!

---

> > > ### Author Response · Authors · 2023-11-18
> > > **Author Response**
> > >
> > > We've added new experiments for ELM using six tasks over Amazon dataset (see Appendix A in the updated version of the paper). Regarding your last comments, we've added captions to the figures as you've proposed (changed marked in red). Thank you again for your review and your positive and helpful comments!

---

> > > > ### Comment · Reviewer_pa1w · 2023-11-18
> > > > **Thank you for the update!**
> > > >
> > > > Thank you for the update! The new experiment results look good. Two minor comments:
> > > >
> > > > 1. It's a good practice to report the compensation provided to the study participants. One way is to report an approximated hourly wage by multiplying the task duration by the task reward. If the rate is found to be below the federal minimum wage, I recommend considering adding bonus reward to the crowd workers.
> > > > 2. Typo: there is a "Google" after "Rohan Anil" in the bibliography on page 10.

---

> > > > > ### Author Response · Authors · 2023-11-21
> > > > > **Author Response**
> > > > >
> > > > > Thank you again for your detailed comments!
> > > > >
> > > > > 1. We completely agree with your point about fairness. While we are not permitted to disclose exact compensation details, our data was generated by paid contractors. They received their standard contracted wage, which is above the living wage in their country of employment. We will also include such a statement in our paper.
> > > > > 2. Thank you for noticing this! We've updated this in the paper.

---

> > > > > > ### Comment · Reviewer_pa1w · 2023-11-21
> > > > > > **Thank you for the update!**
> > > > > >
> > > > > > Got it. Yes, it's important to mention how the raters were recruited (I thought they were Amazon MTurk workers) and if they had been compensated fairly in the paper. I have no further questions. Happy thanksgiving!

---

### Official Review · Reviewer_BRjZ · 2023-11-07

**Soundness:** 2 fair
**Presentation:** 1 poor
**Contribution:** 2 fair
**Rating:** 5
**Confidence:** 4

**Summary:**

This paper presents a paradigm for demystifying the compressed embedding representations of deep models (such as MF trained on the MovieLens dataset in the paper) with Large Language Models (ELM). ELM first generates training data by prompting PaML 2-L with the movie title and task-specific information. The training procedure is composed of two stages: (1) training an adapter to project the domain embedding; and (2) full-training the full model including the language model and the adapter. Then the authors construct different tasks on the MovieLens dataset to test the performance of demystifying domain embeddings.

**Strengths:**

1. It is sound to demystify domain embedding with Large Language Models.
2. The authors introduce their approach step-by-step in Sec 2 & 3, which is clear.
3. The performance is evaluated by human raters.

**Weaknesses:**

1. The training data is generated by PaML 2-L, and ELM uses PaLM 2-XS to interpret the domain embeddings. Their similar architectures and training procedures may make the contribution limited.
2. The authors argue that ELM is a general framework for different tasks, but the experiments only involve one model (MF) on one dataset (MovieLens). Therefore, the generalization of ELM to other models and datasets is not guaranteed.
3. Some related works are overlooked. Existing works [1][2] have shown that tuning the adapter can let LLM demystify and reason over the embeddings of images. The authors should provide more explanations and experiments for their differences.

reference:
* [1] Zhu, Deyao, et al. "Minigpt-4: Enhancing vision-language understanding with advanced large language models." arXiv:2304.10592 (2023).
* [2] Koh, Jing Yu, et al. "Generating images with multimodal language models." arXiv: 2305.17216.

**Questions:**

1. Can ELM perform well with training data generated by other LLMs such as ChatGPT?
2. Can the training data be real other than generated? Some datasets contain descriptions and revives (such as Amazon datasets). Why not use these kinds of datasets?
3. Can ELM perform well with other models and other datasets.

---

> ### Author Response · Authors · 2023-11-14
> **Author Response**
>
> Thank you for your review and your helpful comments. To address your concerns, as you’ve proposed, we are adding new experiments for ELM using the public Amazon product dataset [1], which consists of 9.35M items with textual descriptions, 20.9M users, 233.1M reviews, and 82.83M ratings. Indeed, this dataset contains real data of item descriptions, ratings, and reviews, for which we train ELM on 5-7 tasks (starting with item descriptions, positive reviews, negative reviews, neutral reviews, convincing item summaries, and user profiles). We will update the paper with results for these tasks including human rater evaluations by the end of the week, to leave enough time for the discussion period. We hope this will address your major concerns.
>
> We also note that, while our experiment procedures utilize both PaLM2 models for data generation and embedding interpretation, our method does not require the use of the same model architecture for these procedures. We hope our new experiments on the Amazon dataset will address your concern.
>
> Reference for the Amazon Dataset: \
> [1] McAuley, Julian, Christopher Targett, Qinfeng Shi, and Anton Van Den Hengel. "Image-based recommendations on styles and substitutes." In Proceedings of the 38th international ACM SIGIR conference on research and development in information retrieval, pp. 43-52. 2015.

---

> > ### Author Response · Authors · 2023-11-18
> > **Additional Experiments on the Amazon Dataset**
> >
> > We've added new experiments for ELM using six tasks over Amazon dataset (see Appendix A in the updated version of the paper). We hope these experiments address your main concerns. We hope to use the time left in the discussion period to address any further concerns you may have.a

---

### Author Response · Authors · 2023-11-18
**Additional Experiments on the Amazon Dataset**

Dear reviewers,

Thank you all again for your reviews and your helpful comments.

We've added new experiments using the public Amazon dataset (see Appendix A in the updated version of the paper). We focus on a subset within the "Clothing, Shoes, and Jewelry" category, which involves 5.7 million reviews for 1.5 million products. Similar to the Movielens experiments, we introduced both semantic and behavioral embeddings. Behavioral embeddings were trained on users' rating data, while semantic embeddings were generated using item titles, descriptions, categories, and other meta information. We describe this in detail in Appendix A (all changes marked changes in red).

Our evaluation of ELM includes six tasks: item description, positive review, negative review, neutral review, item endorsements, and user profile. We are still waiting for human rating results for the user profile tasks, but already have comprehensive results of all other tasks in the paper.

Thank you for helping us improve our paper. We hope our new experiments address all reviewers' concerns.

---

> ### Author Response · Authors · 2023-11-21
> **Final Results**
>
> Dear reviewers,
>
> The experiments on the Amazon dataset are now finalized and include human rater results for the user profiles task described above, as well as examples of user profiles. We hope our new experiments have answered all reviewers' concerns. Changes in our revised manuscript are highlighted in red.
>
> We look forward to hearing your follow-up thoughts. We would be very grateful if you would increase your scores if we have addressed your remaining concerns.

---

### Meta-Review · Area_Chair_4gmc · 2023-12-12

**Metareview:**

This paper proposes a new approach (ELM) for interpreting domain-specific embeddings via LLMs. The approach is simple and intuitive: adapter layers are used to project domain embeddings and token embeddings to the same vector space, which allows for the creation of input sequences that mix both domain embeddings and natural language tokens. Using this formulation, the authors create a number of synthetic datasets that require the LLM to produce and reason over rich information about the domain embeddings. The main criticism of the paper (shared by most reviewers) was the limited diversity of the experiments, which in the submitted version was exclusively associated with the MovieLens dataset. The authors added new experiments on the public Amazon reviews dataset that satisfied the reviewers' criticism. Another issue is with the potential difficulty in creating diverse prompts for a new domain. Overall, though, the paper describes an interesting idea and is well executed!

**Justification For Why Not Higher Score:**

I do think the potential difficulty regarding extracting diverse and rich enough data for an arbitrary domain is a limiting factor for this method. It may not generalize beyond review/recommendation settings (obviously these are very important though!)

**Justification For Why Not Lower Score:**

The most serious weakness of the paper (all experiments in one domain) was addressed during the rebuttal period. The paper is interesting and potentially impactful given the popularity of domain embeddings.

---

### Decision · Program_Chairs · 2024-01-16

Accept (poster)